# Inorganic carbon fluxes on the Mackenzie Shelf of the Beaufort Sea

Jacoba Mol[1], Helmuth Thomas[1], Paul G. Myers[2], Xianmin Hu[2], Alfonso Mucci[3]

[1]Department of Oceanography, Dalhousie University, Halifax, B3H 4R2, Canada
[2]Department of Earth and Atmospheric Sciences, University of Alberta, Edmonton, T6G 2E3, Canada
[3]Department of Earth and Planetary Sciences, McGill University, Montreal, H3A 0E8, Canada

*Correspondence to*: Jacoba Mol (jacoba.mol@dal.ca)

**Abstract.** The Mackenzie Shelf in the southeastern Beaufort Sea is a region that has experienced large changes in the past several decades as warming, sea-ice loss, and increased river discharge have altered carbon cycling. Upwelling and downwelling events are common on the shelf, caused by strong, fluctuating along-shore winds, resulting in cross-shelf
Ekman transport, and an alternating estuarine and anti-estuarine circulation. Downwelling carries dissolved inorganic carbon (DIC) and other remineralization products off the shelf and into the deep basin for possible long-term storage in the world oceans. Upwelling carries DIC and nutrient-rich waters from the Pacific-origin upper halocline layer (UHL) onto the shelf. Profiles of DIC and total alkalinity (TA) taken in August and September of 2014 are used to investigate the cycling of carbon on the Mackenzie Shelf. The along-shore transport of water and the cross-shelf transport of DIC are quantified using
velocity field output from a simulation of the Arctic and Northern Hemisphere Atlantic (ANHA4) configuration of the Nucleus of European Modelling of the Ocean (NEMO) framework. A strong upwelling event prior to sampling on the Mackenzie Shelf took place, bringing $CO_2$-rich (elevated $p$CO$_2$) water from the UHL onto the shelf bottom. The maximum on-shelf DIC flux was estimated at 16.9 x $10^3$ mol C d$^{-1}$ m$^{-2}$ during the event. The maximum on-shelf transport of DIC through the upwelling event was found to be 65 ± 15 x $10^{-3}$ Tg C d$^{-1}$. TA and the oxygen isotope ratio of water ($\delta^{18}$O-H$_2$O)
are used to examine water-mass distributions in the study area and to investigate the influence of Pacific Water, Mackenzie River freshwater, and sea-ice melt on carbon dynamics and air-sea fluxes of carbon dioxide ($CO_2$) in the surface mixed layer. Understanding carbon transfer in this seasonally dynamic environment is key to quantify the importance of Arctic shelf regions to the global carbon cycle and provide a basis for understanding how it will respond to the aforementioned climate-induced changes.

## 1 Introduction

Coastal and shelf seas are dynamic areas where the oceanic, terrestrial, and atmospheric carbon reservoirs interact. These areas are influenced by the input of nutrients and carbon from riverine and atmospheric sources as well as by upwelling and physical mixing. Although the total area of continental shelves is small compared to that of the global ocean, elevated levels of primary production and carbon cycling take place, making them globally important for the exchange and storage of
carbon dioxide ($CO_2$) (Gattuso et al., 1998). Shelf seas can act as strong sources or sinks of $CO_2$. The role and contribution

of coastal seas to the global ocean carbon sink is not well constrained and differs strongly between regions (Borges, 2005). The source or sink status of shelf seas in the Arctic varies geographically and throughout the year. In some cases, they have been shown to be a net sink of $CO_2$ (e.g., Bates, 2006; Mucci et al., 2010; Shadwick et al., 2011a; Else et al., 2013; Anderson & Macdonald, 2015), while at other times are shown to be areas of increased upwelling activity and sources of $CO_2$ to the atmosphere (e.g., Williams et al., 2008; Mathis et al., 2012; Pickart et al., 2013b). Quantification of these sources or sinks and understanding the effects on the biogeochemistry of shelf regions, is important to gain a better understanding of how and where carbon is stored and moved in these high latitude areas. Changes in the Arctic are greatly affecting carbon inputs and cycling on the shelves of the Arctic Ocean, and quantifying these changes will lead to a greater understanding of the contribution to the carbon cycle on a global scale. Arctic shelves are expected to be more sensitive than temperate regions to climate change due to faster warming of the shallow water column, the presence, locally, of fewer trophic links (Carmack and Wassmann, 2006), a relatively low alkalinity, and a weak carbonate buffering capacity (Shadwick et al., 2013).

The Mackenzie Shelf is an estuarine environment strongly affected by the Mackenzie River freshwater discharge. The Mackenzie River is the fourth largest source of freshwater and the single largest source of sediment to the Arctic Ocean (Doxaran et al., 2015). Depending on atmospheric and ice forcing, the river plume may be directed northward into the Canada Basin or steered eastward along the coast toward the Canadian Arctic Archipelago (McLaughlin et al., 2011). The delivery of nitrogen (N) with the Mackenzie River discharge has a small to moderate effect on primary production, alleviating N limitation in Beaufort Sea surface waters (Tremblay et al., 2014). This impact is complemented by the upwelling of Pacific Water that provides nutrients to the lower euphotic zone. The Beaufort shelves are subjected to strong seasonal changes, being ice-free in the summer and mostly ice-covered through the winter. A cold, low-density surface layer on the Mackenzie Shelf results from the mixing of river runoff, sea-ice melt, and low salinity Pacific Water, spawning a highly stratified upper water column (Rudels et al., 1996). The anticyclonic Beaufort Gyre exports ice to the south where ice-melt contributes to the freshwater reservoir, enhancing surface stratification (McLaughlin et al., 2011).

Along the shelf-break, a narrow (15-20 km wide), Pacific-borne, nutrient-rich current flows from the Bering Strait east towards the Canadian Arctic Archipelago (Pickart, 2004). Strong Ekman convergence produces downwelling in the Canada Basin and upwelling along the boundary of the Beaufort Sea, contributing to freshwater export to the Beaufort Gyre and upwelling of water from the Arctic upper halocline layer (UHL) onto the shelf (Yang et al., 2006). Wind and ice movements are the dominant controls of circulation on the shelves. Storms can alter the flow of the shelf-break jet and cause upwelling or downwelling on the Beaufort shelves. Arctic storms with westerly winds can accelerate the shelf-break jet and cause downwelling, whereas Pacific storms or a strong Beaufort High pressure cell generate easterly winds that can reverse the jet and create upwelling favourable conditions (Pickart et al., 2013b). Shelf-basin exchanges are promoted by upwelling and downwelling induced by surface stress generated by wind and ice motion (Williams et al., 2006), as well as density-driven plumes through canyons, polynya-forced spreading, and the instability of boundary currents generating eddies (Mathis et al.,

2007). Topographic features such as the Barrow Canyon, the Mackenzie Trough, and the Kugmallit Valley, that interrupt the shelf, are areas of enhanced shelf-break exchanges (Williams et al., 2006; Williams et al., 2008).

Upwelling has been documented on the shelf in the fall when storms are common (e.g. Mathis et al., 2012; Williams and Carmack, 2015; Pickart et al., 2013a) and impacts the shelf region in various ways. Upwelling events provide nutrient and $CO_2$-rich (elevated $p CO_2$) water to the euphotic zone, increasing primary production (Williams and Carmack, 2015) and altering the magnitude and direction of the $CO_2$ flux across the air-sea interface on the shelves of the Beaufort Sea (Mucci et al., 2010; Mathis et al., 2012). Saline water brought onto the shelf in the fall preconditions it for greater salt export through brine drainage in the subsequent winter (Melling, 1993). Upwelling, induced by wind, occurs throughout the year such that the location of the ice edge is important. If the ice edge is beyond the shelf-break, upwelling can bring water from deeper than the shelf-break (80-100 m) up onto the shelf (Carmack and Chapman, 2003) and possibly to the surface where $CO_2$ outgassing can occur. If the ice cover extends over the shelf, upwelling can still occur, but the high $p CO_2$ water remains isolated from the atmosphere and nutrient uptake for biological production is light limited. Upwelling and offshore Ekman transport on the shelf also causes downwelling beyond the shelf-break in the Beaufort Gyre, increasing freshwater content, deepening the nutricline, and decreasing productivity in this area (McLaughlin et al., 2011).

Sustained periods of westerly winds intensify the eastward flowing shelf-break jet and create conditions conducive to the movement of water and materials to the deep basin. Downwelling carries particulate matter and remineralization products off the shelf into the deep basin (Forest et al., 2007), supplying water to the UHL (centred at 100 m depth). The westward flowing arm of the Beaufort Gyre is pushed north further away from the shelf-break by these westerly winds, allowing water from the shelf to flow out beyond this point. Under this scenario, carbon that was sequestered by primary production and sank beneath the surface mixed layer can be transported off the shelf and below the upper halocline in the deep basin. Downwelling events are less frequent than upwelling, acting to offset on-shelf fluxes, but not reversing mixing due to upwelling (Pickart et al., 2013b).

Given changes in sea-ice cover, freshwater input, and wind forcing, carbon cycling on the Beaufort Sea shelves is becoming more important to monitor and understand, as the magnitude of shelf-basin fluxes and the source or sink status of the shelf for $CO_2$ is altered by these changing parameters. In this study, we use measured concentrations of carbonate system parameters and modelled velocity fields to look at the impact of upwelling and downwelling on carbon cycling on the Mackenzie Shelf during August and September of 2014. The cross-shelf transport of DIC is investigated to better constrain carbon transfer on the shelf and its contribution to ocean acidification and $CO_2$ air-sea exchange.

## 2 Methods

### 2.1 Collection and analysis of carbonate parameters

Water samples were collected during an expedition in August and September of 2014 onboard the Canadian Coast Guard Ship (CCGS) *Amundsen*. Samples were taken at the stations shown in Fig. 1 in the Beaufort Sea over the entire water column using a rosette system (24 12-L PVC Niskin bottles) equipped with a Conductivity-Temperature-Depth sensor (CTD, Seabird® SBE 911plus). Water samples for the analysis of dissolved inorganic carbon (DIC) and alkalinity (TA) were drawn directly from the Niskin bottles into 300 mL borosilicate glass bottles into which 20μL of a supersaturated mercuric chloride ($HgCl_2$) solution was injected to halt biological activity before being sealed with ground-glass stoppers and Apiezon® Type-M high-vacuum grease. Samples were stored at 4°C in the dark until analysis using a VINDTA 3C (Versatile Instrument for the Determination of Titration Alkalinity, by Marianda) at Dalhousie University following the methods described in Dickson et al. (2007). The instrument was calibrated against Certified Reference Materials provided by A.G. Dickson (Scripps Institute of Oceanography) and the reproducibility of the DIC and TA measurements was, respectively, better than 2 μmol kg$^{-1}$ and 3 μmol kg$^{-1}$. Water samples for measurements of the carbon isotope ratio of DIC ($\delta^{13}$C-DIC) and the oxygen isotope ratio of water ($\delta^{18}$O-$H_2$O) were collected in tandem with DIC/TA samples. $\delta^{13}$C-DIC samples were analyzed at the Yale Analytical and Stable Isotope Center (Yale University) using a GasBench II system connected to a Thermo Delta Plus XP isotope ratio mass spectrometer (IRMS). $\delta^{18}$O-$H_2$O samples were analyzed on a triple collector IRMS in dual inlet mode at the GEOTOP Stable Isotope Laboratory (Université du Québec à Montréal). The $p$CO$_2$, pH (on the total scale), and aragonite saturation state ($\Omega_{Ar}$) were computed using DIC and TA as input parameters with the standard set of carbonate system equations, excluding nutrients, using the CO$_2$SYS program of Lewis and Wallace (1998) and the carbonic acid dissociation constants of Mehrbach et al. (1973) refit by Dickson and Millero (1987).

### 2.2 Water mass definitions

The distribution and vertical structure of water masses in the Arctic Ocean are well known and documented (e.g., Jones and Anderson, 1986; Aagaard, 1989; Swift et al., 1997; Jones et al., 1998). The variable inputs of freshwater from rivers and sea-ice melt have a strong impact on the local stratification and circulation (e.g., Aagaard and Carmack, 1989; Rabe et al., 2011; McClelland et al., 2012). In this study, the vertical water column is divided into three major water masses. At the surface is the polar mixed layer (PML), a mixture of meteoric/river water (MW), sea-ice melt (SIM), and upper halocline Pacific-origin water. The UHL originates in the Pacific Ocean and lies below the PML. It covers a depth range of approximately 25 to 200 m, a salinity range of 31.6 to 34.6 (with a core salinity of 33.1), and is characterized by a temperature minimum. Below the UHL is the Atlantic layer (ATL), covering the depths below 150 m to approximately 1000 m, with a salinity range of 34.6 to 34.9. Waters with a mixture of UHL and ATL properties, or without at least 80 % of the total fraction of one of these water masses, are defined as UHL-ATL waters. This described layering of water masses is applicable over the deep basin in the study area and disappears somewhat over the shelf as mixing and upwelling/downwelling dynamics occur.

Using a three end-member mixing scheme allowing for negative values, assuming conservative mixing, and knowing the TA and $\delta^{18}$O-H$_2$O of each water mass, the relative fraction of each water mass in a sample of known TA and $\delta^{18}$O can be calculated. This technique has been applied successfully in multiple independent studies in the same geographic area (e.g., Yamamoto-Kawai et al., 2009; Shadwick et al., 2011b). The upper 150 m of the water column is assumed to be a mixture of MW, SIM, and UHL, and the following system of linear equations is solved to estimate the relative fraction of each water mass:

$$f_{MW} + f_{SIM} + f_{UHL} = 1 \tag{1}$$
$$TA_{MW}f_{MW} + TA_{SIM}f_{SIM} + TA_{UHL}f_{UHL} = TA \tag{2}$$
$$\delta^{18}O_{MW}f_{MW} + \delta^{18}O_{SIM}f_{SIM} + \delta^{18}O_{UHL}f_{UHL} = \delta^{18}O \tag{3}$$

where $f_x$ represents the relative fraction of each water mass. Below 150 m, the water is assumed to be a mixture of net SIM, UHL, and ATL, and the following system of equations is used:

$$f_{SIM} + f_{UHL} + f_{ATL} = 1 \tag{4}$$
$$TA_{SIM}f_{SIM} + TA_{UHL}f_{UHL} + TA_{ATL}f_{ATL} = TA \tag{5}$$
$$\delta^{18}O_{SIM}f_{SIM} + \delta^{18}O_{UHL}f_{UHL} + \delta^{18}O_{ATL}f_{ATL} = \delta^{18}O \tag{6}$$

The end-member values used for each water mass are shown in Table 1. The TA value for MW is a flow-weighted value of the Mackenzie River water taken from Cooper et al. (2008). The $\delta^{18}$O value for MW is a flux-weighted value for the Mackenzie River taken from Yi et al. (2012). These endmembers give a balanced value for the river water endmember throughout the year. Although Alaskan and Eurasian rivers may have some influence on the upper water column in this area and contribute water with different properties, the dominant input of MW to the study area is the Mackenzie River and so these values are used. The value of DIC for MW is taken from Shadwick et al. (2011b). The value of $\delta^{18}$O for SIM is taken from Yamamoto-Kawai et al. (2009) and the values of TA, DIC and salinity for SIM are taken from Lansard et al. (2012). Values of $\delta^{18}$O, TA and DIC for the UHL are the average values of all water samples at a salinity of 33.1, considered to be the core of the UHL water mass. Likewise, ATL values are averages of all values at the deep temperature maximum and a salinity of 34.8.

**2.3 Arctic and Northern Hemisphere Atlantic Simulation**

The velocity fields used in this study are taken from the ¼ degree Arctic and Northern Hemisphere Atlantic (ANHA4) configuration. This is a regional configuration of a coupled ocean-sea ice model, based on the Nucleus for European Modelling of the Ocean (NEMO, Version 3.4) framework (Madec, 2008). The sea-ice model used is the Louvain-La-Neuve

Sea-Ice Model Version 2 (LIM2) with an elastic-viscous-plastic (EVP) rheology, including both thermodynamic and dynamic modules (Fichefet and Maqueda, 1997). The horizontal mesh is a subdomain of the global ORCA025 tripolar grid with two open boundaries, one close to the Bering Strait in the Pacific Ocean and the other at 20° S in the Atlantic Ocean. In the vertical, there are 50 unequally spaced geopotential levels with higher resolution (~ 1 m) in the top 10 m.

This simulation is integrated from January 2002 to December 2014. The initial conditions (3D temperature, salinity, horizontal velocities, sea surface height and sea ice) are extracted from the Global Ocean Reanalysis and Simulations (GLORYS2v3) produced by Mercator Ocean (Masina et al., 2015). Hourly, 33 km horizontal resolution atmospheric forcing data (10 m wind, 2 m air temperature and humidity, downwelling and longwave radiation fluxes, and total precipitation) from the Canadian Meteorological Centre (CMC) global deterministic prediction system (GDPS) reforecasts (CGRF) dataset (Smith et al., 2014) are used to drive the model. The GLORYS2 dataset is also used to provide the open boundaries (temperature, salinity and ocean velocities). Monthly inter-annual runoff from Dai et al. (2009) as well as Greenland meltwater provided by Bamber et al. (2012) are also carefully remapped onto the model grid to give a more realistic freshwater input from the land to the ocean.

Model output is 5-day averages of the velocity fields in the study area for August and September of 2014. To look at along-shore and cross-shelf transports on the Mackenzie Shelf, the velocity was gridded along the shelf-to-basin sampled transect (Fig. 1) every 5 km, starting from 20 km closer to shore than the first sampled station and ending ~ 20 km into the basin beyond the last sampled station. The along-shore direction runs parallel to the shelf-break, at a bearing of 52° to the east of true north (52° T or 232° T). The cross-shelf direction is then 38° to the west of true north (142° T or 322° T) (directions indicated in Fig. 1). The ocean current vectors (u and v components) from the ANHA4 simulation were tilted to correspond to these along-shore and cross-shelf bearings, resulting in the u-component of the velocity representing along-shore flow and the v-component representing cross-shelf flow. In all cases, positive (negative) values indicate along-shore flow to the east (west) and cross-shelf flow in the off-shelf (on-shelf) direction.

### 2.4 Flux calculation uncertainty

Uncertainty was estimated using Monte Carlo simulations for the calculation of cross-shelf mass transports. The inputs for the simulations were randomly generated from normal distributions, and 1000 points were chosen randomly for each of the variables. Standard deviation for salinity (± 0.0003) and temperature (± 0.001 °C) were taken from error calculations with the Seabird 911plus CTD sensor, and the standard deviation for DIC measurements (± 2 µmol kg$^{-1}$) was taken from repeat measurements of sample and standard seawater using the VINDTA 3C. The error of the modelled velocity was estimated by taking the standard deviation of the velocity at each station and depth layer from the ANHA4 model over the two-month study period.

# 3 Results

## 3.1 Carbonate system

High variability in the carbonate system properties of surface waters is observed throughout the study region with marked differences between the basin and distinct shelf regions (Fig. 2). Temperature varies from less than -1 °C in the central basin and beyond the shelf break to greater than 6 °C on the Mackenzie Shelf and in Amundsen Gulf. Salinity ranges from less than 26 to greater than 30, with the lowest values in areas close to the Mackenzie River and in the central basin. Higher salinity values in Amundsen Gulf and on the Mackenzie Shelf suggest upwelling of high salinity deep water. DIC is lowest in the low-salinity and low-temperature surface waters of the central basin, and higher in shelf seas, possibly a result of upwelling. TA follows a pattern similar to that of salinity, with low values beyond the shelf break and in the central basin, and higher values over the shelf. High $p$CO$_2$ values are found in waters with higher temperature and DIC content, primarily over the shelf and in Amundsen Gulf. Lower $\delta^{13}$C-DIC values are observed over the shelf in areas near the Mackenzie River outflow, as river-water values are presumably depleted due to accumulation of metabolic CO$_2$ from the microbial respiration of terrigenous or riverine organic matter, while high $\delta^{13}$C-DIC values are found in the northern Amundsen Gulf.

These surface plots (Fig. 2) are constructed from surface samples taken over more than a month-long period, from August 17[th] to September 21[st]. The stations along each of the three transects, Amundsen Gulf, Mackenzie Shelf and Mackenzie Trough, were all sampled within a three-day span, except for the furthest off-shore station on the Mackenzie Trough transect. Temporal variability between the three transects and the stations in the deep basin must be considered when looking at the spatial variability between these areas. Changes in wind and circulation that occurred through the sampling period may alter the surface conditions due to upwelling or downwelling, where the Mackenzie River discharge is directed, and concentrations of sea-ice in different areas.

Relationships between DIC, TA, salinity, temperature, $p$CO$_2$ and $\Omega_{Ar}$ for all samples on the Mackenzie Shelf, in Amundsen Gulf and in the Canada Basin are shown in Fig. 3. DIC and TA are both quite variable in the low salinity PML with freshwater from sea-ice melt and river outflow contributing waters of different chemical properties (Fig. 3a and b). DIC ranges widely from 1822 to 2156 µmol kg$^{-1}$ in the PML and TA ranges from 1892 to 2269 µmol kg$^{-1}$ over this same large salinity range of 25.8 to 32.0. DIC reaches a maximum of $\sim$ 2225 µmol kg$^{-1}$ in samples characterized by the highest relative contributions of UHL, a common feature in this study area. For example, Shadwick et al. (2011b) found a DIC maximum of 2240 µmol kg$^{-1}$ at the UHL maximum, while Brown et al. (2016) found a DIC maximum of 2225 $\pm$ 8 µmol kg$^{-1}$, and Lansard et al. (2012) a DIC maximum of 2228 $\pm$ 8 µmol kg$^{-1}$ in this same water mass. TA shows a maximum of $\sim$ 2360 µmol kg$^{-1}$ in waters with the greatest ATL contribution and a salinity maximum of $\sim$ 34.9. These parameters follow the trends seen in previous years in the same sampling areas (e.g. Shadwick et al., 2011b). A linear regression was performed to extrapolate to the values of DIC and TA at a salinity of zero, representative of the Mackenzie River water end-member (regression lines

shown in Fig. 3a and b). Only samples with greater than 10 % $f_{MW}$ and less than 5 % $f_{SIM}$ were used for the regression. The resulting end-member values of DIC and TA for the Mackenzie River are 1441 µmol kg$^{-1}$ and 1564 µmol kg$^{-1}$ respectively. These values are close to the Mackenzie River properties published in Shadwick et al. (2011b) and Cooper et al. (2008) and used for the water mass deconvolution in this study (Table 1).

The relationship between DIC and TA is shown in Fig. 3c. For samples in the PML the slope is < 1, with TA increasing at a faster rate than DIC. TA is nearly conservative while DIC is more responsive to biological processes, including photosynthesis and respiration in the PML. The slope of this relationship increases in the UHL to > 1, as DIC builds up in the mid-depth layer to its maximum value due to the accumulation of remineralization products. A greater DIC:TA ratio
indicates that there is a greater amount of DIC in the water relative to TA, altering the chemistry and leading to an increase in $p$CO$_2$. Values of the DIC:TA ratio for the core or average values of each of the water masses defined here are shown in Table 1. $p$CO$_2$ is variable in the PML, ranging from ~ 200 to 500 µatm, but mostly staying at values below 400 µatm (Fig. 3d). Temperatures range from -2 °C to 7 °C in the PML, with resulting variability in $p$CO$_2$ and $\Omega_{Ar}$ (Fig. 3e and f). The UHL is where $p$CO$_2$ reaches its maximum, with values greater than 700 µatm in water associated with the DIC maximum. The
UHL is also the layer where the temperature minimum is found (Fig. 3e and f). The high values of $p$CO$_2$ and low $\Omega_{Ar}$ found in the UHL are restricted to a small temperature range. Waters return to the 200 to 500 µatm range in the ATL layer, with a few samples showing higher values.

Five stations were sampled along an onshore-offshore transect across the Mackenzie Shelf and beyond the shelf-break from
August 22[nd] to 24[th] (identified in Fig. 1). Water properties and carbonate system parameters in the top 300 m are shown in Fig. 4. Water temperature is warmest at the surface over the shelf, reaching values of > 6.5 °C. Surface waters over the deep basin are cooler, with a maximum temperature of < 1 °C. Salinity, DIC and TA all show similar patterns in the top 300 m over the shelf and beyond the shelf-break. The lowest values (< 26.5, 1830 µmol kg$^{-1}$ and 1920 µmol kg$^{-1}$ for salinity, DIC and TA respectively) are seen in the surface water beyond the shelf-break. Salinity and TA increase down to 300 m, reaching
maximum values of ~ 34.8 and ~ 2300 µmol kg$^{-1}$ respectively. DIC has a mid-depth maximum at around 125 m depth of ~ 2225 µmol kg$^{-1}$, then decreases slightly with depth down to 300 m. All three of these parameters show isoclines sloping up onto the shelf, with the same values at shallower depths over the shelf, as well as elevated surface values over the shelf.

Low $p$CO$_2$ values are observed in the surface layer along the transect, ranging from 278 to 339 µatm. Beyond the shelf-
break, $p$CO$_2$ increases with depth to a maximum value of 749 µatm at around 125 m depth in the UHL (Fig. 4e). Below this depth, the $p$CO$_2$ decreases gradually. Maximum values on the shelf of up to 665 µatm are found in the bottom layer (53 m depth). Measurements of δ$^{13}$C-DIC show the expected pattern, with highest values of up to 2.20 ‰ in the surface or subsurface, due to photosynthesis and the preferential uptake of the lighter carbon isotope, and a decrease to minimum values of -0.5 ‰ in the DIC maximum layer in the deep basin and on the shelf bottom due to respiration of $^{13}$C-depleted

organic matter. Maximum values of $pCO_2$ and minimum values of $\delta^{13}$C-DIC are coincident on the shelf bottom and at depths of around 125 m beyond the shelf-break. The temperature effect on $pCO_2$ is not overly important at these depths, as large variations in temperatures are most prominent in the top 30 m (Fig. 4a). Variations in $pCO_2$ at these greater depths can be attributed to changes in DIC content (metabolic $CO_2$ addition), as indicated by the depleted $\delta^{13}$C-DIC values at the same

locations. The saturation state of the waters with respect to aragonite ($\Omega_{Ar}$) and pH are also closely related to the DIC concentration and $pCO_2$ levels, with the lowest values being observed along the shelf bottom and in the DIC maximum beyond the shelf-break (Fig. 4g and h). These parameters and the consequences for ocean acidification are discussed further in Sect. 4.4.

### 3.2 Water mass composition

Surface water composition varies greatly over the study region with large freshwater input from the Mackenzie River as well as significant sea-ice melt in the central Canada Basin (Fig. 5a and b). The strong impact of the Mackenzie River is seen easily, with fractions of meteoric water greater than 20 % observed over the shelf. The river plume is detectable (> 10%) in the surface waters to the east into Amundsen Gulf and far into the central Canada Basin. Sea-ice melt has the strongest influence in the central basin with fractions of ~ 15 %, and substantial fractions of up to 10 % also seen near the shelf break.

The fraction of water-mass types along the sampled transect on the Mackenzie Shelf are shown in Fig 5c-f. High meteoric water content is seen over the shallow shelf region, as well as in the surface layer out into the deep basin. There is a notable intrusion of sea-ice melt water from the deep basin onto the shelf break on the Mackenzie Shelf. This sea ice may be transported south on the eastern arm of the Beaufort Gyre and pushed onto the shelf. The UHL dominates at mid-depth,

around 100 m, off the shelf, with large fractions (> 95 %) pushed up to shallower depths along the bottom of the shelf. Atlantic water is restricted to the deep layer beyond the shelf break and did not intrude up onto the shallow shelf during our sampling period.

### 3.3 Wind forcing

Wind data was taken from the gridded reforecast data of the Canadian Meteorological Centre (Smith et al., 2014). The

location selected is on the Mackenzie Shelf, near the middle of the transect located at 70.3125° N, 133.5937° W (Fig. 1). Figure 6 shows the wind direction and magnitude averaged every six hours through August and September of 2014. Oscillations between strong northerly and southerly winds are a prevalent feature. From August 6[th] to 16[th], the wind field was dominated by northeasterly winds with magnitudes of > 20 m s[-1]. Between August 26[th] and September 10[th], periods of strong southerly winds were interrupted by short periods of strong northerly winds with maximum velocities of > 50 m s[-1] in

both directions. September 14[th] to 16[th] and 26[th] to 30[th] were periods of sustained southeasterly winds reaching speeds of up to 40 m s[-1]. Sampling of the Mackenzie Shelf transect took place from August 22[nd] to 24[th], indicated by the red lines in Fig. 6.

During this time, the wind direction was variable and wind speeds were comparably low. This sampling period followed both northeasterly and southeasterly winds throughout the beginning of the month.

### 3.4 Mackenzie Shelf circulation

The modelled velocity, averaged in the top 10 m of the water column, on the Mackenzie Shelf is shown in Fig. 7a and b for two time periods displaying the two opposite modes of circulation observed during the study period in the fall of 2014. During the period from August 16th to 20th, surface water flows to the west along the Mackenzie Shelf. From September 5th to 9th, mean surface flow is towards the east. In the middle of the shelf, cross-shelf transport of surface water is evident at both times, with off-shelf flow from August 16th to 20th and on-shelf flow from September 5th to 9th. In both cases, the Cape Bathurst topography induces intensified surface currents in the mean direction of flow. Surface flow is also altered at the Mackenzie Trough, as topography becomes complex and the influence of the Mackenzie River is greatest. The modelled velocity at the 56 m depth horizon shows the strong influence of the shelf-break jet from both August 16th to 20th (Fig. 7c) going to the west and from September 5th to 9th (Fig. 7d) to the east. The water flow closely follows the topography of the shelf in both cases, shown clearly at the Mackenzie Trough, but also evident with cross-shelf movement through the middle of the sampled transect, most likely due to bathymetric changes at the Kugmallit Valley, a previously documented location of enhanced cross-shelf transport (Williams et al., 2008). Cross-shelf movement is also seen at several other locations along the shelf-break. From August 16th to 20th, the modelled velocity at the 92 m depth horizon shows an accelerated shelf-break jet (Fig. 7e). The velocity from September 5th to 9th at the 92 m depth horizon (Fig. 7f) shows the eastward flowing shelf-break jet of similar velocity as at 56 m depth.

Two opposite modes of circulation are shown to occur on the Mackenzie Shelf during the study period. From August 16th to 20th, the shelf-break jet flows to the west whereas, from September 5th to 9th, it flows to the east. In the former case, it generates conditions conducive to upwelling along the shelf and produces an estuarine circulation, with relatively fresher water in the surface layer pushed off-shelf towards the basin and water at depth being brought onto the shelf. In the latter case, downwelling could take place, producing an anti-estuarine circulation, with water at the surface moving shoreward and denser water at depth moving out past the shelf-break. These two periods are investigated further to look at the upwelling and downwelling circulation in the fall of 2014.

### 3.5 Velocity fields

The two-month average, modelled velocities of the along-shore and cross-shelf flow for the Mackenzie Shelf transect are shown in Fig. 8a and b. The surface flow over the shelf is dominated by eastward along-shore flow. The westward flow of the Beaufort Gyre beyond the shelf-break is evident. The cross-shelf flow is primarily in the on-shelf direction over the shallow shelf and off-shelf beyond the shelf-break. Due to Ekman transport, westward flow towards the Alaskan Shelf is associated with upwelling onto the shelf. This situation is seen from August 16th to 20th with strong westward along-shore

transport and velocities on the shelf and in the core of the shelf-break of > 0.15 m s$^{-1}$ (Fig. 8c). This westward transport throughout the water column coincides with water velocities in the cross-shelf direction indicative of upwelling (Fig. 8d). Surface water over the shelf reaches velocities of > 0.08 m s$^{-1}$ in the off-shelf direction in the top 5 m and flow remains in an off-shelf direction for much of the top 15 m of the water column. Below this surface layer, water flow over the shelf is in an

on-shelf direction, reaching velocities of > 0.10 m s$^{-1}$ at mid-depth ($\sim$ 50 m). This upwelling flow is readily noted, with water at depth being moved onto the shelf and water at the surface moving out toward the shelf-break. Flow from the west towards Amundsen Gulf is associated with downwelling transport off the shelf, a situation seen from September 5$^{th}$ to 9$^{th}$ (Fig. 8e and f). Along-shore velocity over the shelf and shelf-break is high, dominated by flow in the eastward direction over the entire shelf area, reaching values of > 0.16 m s$^{-1}$ at the surface and decreasing with depth. The westward flow in the deep basin

decreases and is pushed further beyond the shelf-break relative to the average along-shore pattern. The strong eastward flow over the shelf and shelf-break is associated with a small on-shelf transport at the surface (> 0.09 m s$^{-1}$) and an off-shelf transport along the shelf bottom reaching velocities of > 0.03 m s$^{-1}$. This cross-shelf velocity pattern shows a downwelling loop, with water at the surface pushed shoreward and water at depth moving out toward the shelf-break.

These modelled velocities are representative of 5-day averages for the region and thus do not show the peak velocities reached during the study period. The shelf-break jet or boundary current has been measured throughout the year at various locations along the Alaskan and Canadian Beaufort shelves. For example, Pickart et al. (2009) reported a bottom-intensified shelf-break jet flowing at $\sim$ 0.15 m s$^{-1}$ to the east over a year-long mean across the shelf at 152°W, as similarly reported by Nikolopoulos et al. (2009). This value is the same as the maximum modelled velocities found in the core of the shelf-break

jet through the sampled transect. Reversals of the shelf-break jet due to variations in wind forcing have been reported in numerous studies along the western and eastern Beaufort Shelf (e.g. Nikolopoulos et al., 2009; Pickart et al., 2013b; Dmitrenko et al., 2016). Measurements of velocity from moorings anchored in the same region across the Mackenzie Shelf slope, such as those analyzed in Forest et al. (2015) from September 2009 to August 2012, show current surges of 0.20 to 0.80 m s$^{-1}$ with frequent oscillations in current direction. The mean velocities over this period at these moorings were 0.08 to

0.14 m s$^{-1}$. These mean velocities are comparable in magnitude to velocities produced by the model for 5-day averages. It is likely that similarly strong surges in current velocity took place throughout the study period of August and September 2014 on the Mackenzie Shelf.

## 4 Discussion

### 4.1 Along-shore volume transport

Along-shore water transport across the Mackenzie Shelf transect throughout the study period was estimated in the top 200 m of the water column through the 120 km (in the cross-shelf direction) section identified in Fig. 1 (Fig. 9a). Transport was calculated by taking the product of the depth-averaged modelled velocity and the depth and cross-shelf area between

stations. Positive values indicate eastward flow, whereas negative values indicate westward flow. Maximum westward transport of 0.74 Sv through the Mackenzie Shelf transect took place from August 16[th] to 20[th]; maximum eastward transport of 0.89 Sv through the transect occurred from September 5[th] to 9[th]. In comparison, the transport of northward flowing water through Bering Strait is approximately 1.1 Sv (Woodgate et al., 2012). The along-shelf transport through these two transects
and the changes in transport direction are substantial.

In August and September of 2014, the circulation along-shore changed from a strong flow to the west to a strong flow to the east and back again (Fig. 9). Correlation between along-shore wind in the northeasterly (southwesterly) direction and upwelling (downwelling) on the Mackenzie Shelf has been shown in numerous studies and models (e.g. Carmack &
Kulikov, 1998; Yang et al., 2006; Mathis et al., 2012). In our study, the two largest upwelling episodes on the Mackenzie Shelf, August 11[th] to 20[th] and September 15[th] to 24[th], were preceded by periods of strong ($> 20$ m s$^{-1}$) and sustained northeasterly winds (Fig. 6). The largest downwelling episodes, from August 31[st] to September 9[th], appear to be triggered by strong southwesterly winds ($> 40$ m s$^{-1}$) and strong wind reversals throughout.

The oscillating along-shore water transport is dominated by changes over the slope, as evidenced by the transport calculated at the specific sampled station locations. Along-shore transport at the stations on the transect are calculated through the water column, to a maximum depth of 200 m, by taking the sum of the products of the along-shore modelled velocity and a cross-shelf distance of 1 km (Fig. 9b). Bottom depth was taken from the shipboard depth sounder when sampling took place. All three stations on the shelf (434, 432 and 428) as well as station 435 on the slope show the same oscillating change in along-
shore transport. Station 421, located beyond the slope in the deep Canada Basin shows a negative total along-shore transport throughout the study period. This station is under the influence of the Beaufort Gyre, which is consistently flowing in a westward direction at this location. During the 2-month study period there is an oscillation between two dominant flow patterns represented by positive and negative total transport in the along-shore direction.

**4.2 Cross-shelf carbon transport**

Cross-shelf flux calculations were carried out for the Mackenzie Shelf transect to investigate the exchange of DIC and TA between the shelf and the deep basin. Fluxes were calculated by taking the product of linearly gridded DIC or TA measurements and cross-shelf velocity values from the ANHA4 simulation through the water column with an along-shore distance of 10 km. The DIC flux (mol C d$^{-1}$ m$^{-2}$) at the shelf stations for the period of the largest on-shelf flux are shown in Fig. 10. From August 16[th] to 20[th], during the period of greatest westward along-shore transport, the cross-shelf DIC flux is
off-shore at all stations in the top 10 to 25 m of the water column, reaching values of 16.7 x 10$^3$ mol C d$^{-1}$ m$^{-2}$ (TA flux: 18.1 x 10$^3$ mol TA d$^{-1}$ m$^{-2}$) at station 432. The DIC flux is on-shelf below the surface layer, with stations 434 and 432 having maximum on-shelf fluxes at mid-depth. At station 428, the on-shelf flux reaches values of 16.9 x 10$^3$ mol C d$^{-1}$ m$^{-2}$ (TA flux:

$17.4 \times 10^3$ mol TA $d^{-1}$ $m^{-2}$) near the bottom of the water column. This on-shelf flux near the ocean floor transports water sourced from the UHL, bringing metabolite-rich water and DIC onto the shelf.

The excess DIC brought onto the shelf and its impact on the $p\text{CO}_2$ and pH depends on the amount and composition of the source-water. The DIC:TA ratio in the water being moved along the bottom of the shelf is a parameter that provides insight into the carbonate system properties of the water mass. At stations 434, 432 and 428 on the shelf, the DIC:TA ratio changes significantly with depth. Water being upwelled (Fig. 10, negative DIC flux) has DIC:TA ratios greater than 0.975 at all three stations. In other words, excess DIC is being upwelled, contributing to increased $p\text{CO}_2$ and decreased pH and $\Omega_{Ar}$ levels of the shelf bottom waters. The DIC:TA ratio in the latter is higher than the ratio in the surface waters (0.92 to 0.93) being moved out towards the Canada Basin. During the time of measurement, surface $p\text{CO}_2$ levels along the Mackenzie Shelf transect remained low, ranging from 278 to 339 µatm, allowing for the uptake of $\text{CO}_2$ by the surface ocean.

Cross-shelf mass transport of carbon was calculated for the three shelf stations (434, 432 and 428) from August 16th to 20th (Table 2). These transports are integrated through a 10 km along-shore distance to include the area of the shelf influenced by similar water movement. From August 16th to 20th, when the along-shore flow was westward, the cross-shelf transport should have lead to an upwelling flow over the shelf. As expected by the cross-shelf velocity in Fig. 8d, transport in the top 10 m at all three stations is off-shelf, reaching a maximum of $14 \pm 3 \times 10^{-3}$ Tg C $d^{-1}$. Below the surface layer, the cross-shelf transport is on-shelf, with maximum transport at station 428 of $65 \pm 15 \times 10^{-3}$ Tg C $d^{-1}$.

From August 6th to 16th, strong northeasterly winds were present over the shelf region, but had shifted to an easterly direction by the time sampling took place and through August 22nd. Although no results from a coupled physical-biogeochemical model were available, it seems probable from the wind and physical sampling data that upwelling on the Mackenzie Shelf began around August 6th and continued through to the 22nd of August. From the analysis of the hydrodynamic model, there are two periods, lasting longer than 10 days in August and September of 2014 (approximately August 10th to 28th and September 14th to 29th), during which the cross-shelf transport brings water from beyond the shelf-break onto the Mackenzie Shelf. During these episodes, water from the DIC-rich UHL is brought onto the Mackenzie Shelf. The period of off-shelf transport lasts for ~ 15 days in the region, offsetting the on-shelf flux of these waters and carrying remineralization products (metabolites) back to the deep basin. Nevertheless, the average transport through the two months carries water onto the shelf. This long-term upwelling activity acidifies the shelf bottom waters and may impact the air-sea exchange of $\text{CO}_2$ due to the large air-sea $p\text{CO}_2$ difference, including altering the sink or source status of the shelf. Surface waters with high meteoric water content are delivered beyond the shelf-break into the Beaufort Gyre as upwelling takes place.

**4.3 Impacts of upwelling on carbonate chemistry**

The result of upwelling on the shelf is reflected by gradients of DIC, TA and $\delta^{13}$C-DIC across the shelf (Fig. 11a-d). The on-shore transport of deeper water is evident when comparing the values at the 50 m depth horizon across the three identified stations. The onshore gradients for DIC and TA are -1.98 $\mu$mol kg$^{-1}$ m$^{-1}$ and -0.87 $\mu$mol kg$^{-1}$ m$^{-1}$ respectively. The DIC:TA

ratio of these gradients is 2.28, meaning that more DIC, relative to TA, is brought onto the shelf. This is important because a higher DIC:TA ratio translates into an increased $p$CO$_2$ of the water as well as a decrease in the pH and the saturation state of the water with respect to carbonate minerals. If these waters are brought to the surface by upwelling or vertical mixing, the net flux of CO$_2$ over the shelf region may be altered. Outgassing may occur, releasing CO$_2$ to the atmosphere, or the uptake of CO$_2$ by the surface ocean may be suppressed. Although surface $p$CO$_2$ was undersaturated with respect to the atmosphere

when sampling took place, outgassing or a bloom may have occurred before sampling at the height of this upwelling period. The gradient of $\delta^{13}$C-DIC across the shelf is positive (Fig. 11c and d), opposite to that of DIC and TA. $\delta^{13}$C-DIC values reach minima along the shelf bottom, reflecting the signature of UHL source water and remineralization products on the shelf.

To investigate the delivery of DIC to the shallow shelf, the effects on the source or sink status of the shelf, and the pH conditions, the excess DIC (DIC$_{ex}$) and excess protons (H$^+_{ex}$) at 50 m depth were calculated relative to those of a parcel of water of identical TA, temperature, and salinity in equilibrium with the atmosphere at a $p$CO$_2$ of 400 $\mu$atm (following the methods of Burt et al., 2013):

$$DIC_{ex} = DIC_{obs} - DIC_{(pCO_2=400)} \tag{7}$$
$$H^+{}_{ex} = H^+{}_{obs} - H^+{}_{(pCO_2=400)} \tag{8}$$

The gradient in DIC$_{ex}$ is -1.19 $\mu$mol kg$^{-1}$ km$^{-1}$ and the gradient in H$^+_{ex}$ is -0.09 nmol kg$^{-1}$ km$^{-1}$ (Fig. 11e). Both quantities display negative gradients, with positive values closer to shore along the shelf bottom at station 432 where DIC$_{ex}$ reaches a

value of 51.4 $\mu$mol kg$^{-1}$ and H$^+_{ex}$ reaches a value of 4.57 nmol kg$^{-1}$. If this shelf water is upwelled, the surface water $p$CO$_2$ and pH conditions will be altered, lowering the pH and raising the $p$CO$_2$ at the surface leading to outgassing. At station 432 there is excess DIC in the water column and an on-shore water movement from 35 to 55 m depth. The total on-shore cross-shelf mass transport of DIC$_{ex}$ (relative to a $p$CO$_2$ of 400 $\mu$atm) during the upwelling event from August 16$^{th}$ to 20$^{th}$ at the three shelf stations is shown in the last column of Table 2.

**4.4 Ocean acidification**

Distinguishing the ocean acidification signal in the Arctic is complicated because of the interplay of several changing environmental conditions including warming, sea-ice loss, surface freshening and changes in primary production (Carmack

et al., 2016). During sampling, a layer of high $pCO_2$, low pH water resided at depth on the shelf (Fig. 4e and h), having been upwelled from beyond the shelf-break onto the shelf. This bottom layer is sourced from the UHL where a DIC maximum consistently resides beyond the shelf-break (Anderson et al., 2010; Yamamoto-Kawai et al., 2013). The UHL source-water is of Pacific-origin, and is preconditioned before entering the Arctic to have low $\Omega_{Ar}$ values (Yamamoto-Kawai et al., 2013).

This water is seen in Fig. 3e and f, centred at a salinity of 33.1 and a temperature minimum, as having $pCO_2$ values over 600 µatm and $\Omega_{Ar}$ as low as 0.83. This water can be corrosive to the mineral skeletons and shells of $CaCO_3$-secreting organisms such as bivalves, mollusks, and echinoderms, with varying impacts on different organisms (Ries et al., 2011). These conditions can also have negative impacts on community structure and recruitment that may carry through to higher trophic levels (Fabry et al., 2008). In our study area, the undersaturated water mass is not at the surface over the shelf, but sustained

periods of upwelling could bring this high $pCO_2$ water to the surface and outgas $CO_2$ to the atmosphere (e.g. Mucci et al., 2010; Mathis et al., 2012). An outgassing event following wind-induced upwelling on the Alaskan Beaufort Shelf was documented by Mathis et al. (2012). This was a 10-day event that brought water of $\Omega_{Ar} < 1.0$ to the surface over the Beaufort shelf. Likewise, upwelling of elevated $pCO_2$ (522 ± 37 µatm) UHL water at Cape Bathurst was captured by Mucci et al. (2010). In the present study, sampling took place just after a suspected strong upwelling event. Aragonite undersaturated

waters with high $pCO_2$ were restricted to depths of greater than 20 m over the shelf at the time of sampling. A strong upwelling wind-event could easily have brought this water to the surface and created events like those described by Mucci et al. (2010) and Mathis et al. (2012).

## 5 Conclusions

Carbonate system parameters were measured in the Beaufort Sea during August and September of 2014. The cross-shelf

transport of DIC during an upwelling event was estimated using the velocity fields from output of the ANHA4 simulation. The upwelling period from August 16th to 20th displayed an off-shelf transport in the surface layer of $14 \pm 3 \times 10^{-3}$ Tg C d$^{-1}$ and a corresponding on-shelf transport in the subsurface of $65 \pm 15 \times 10^{-3}$ Tg C d$^{-1}$, bringing water from the UHL onto the shelf. The upwelled UHL water alters the carbonate chemistry of bottom waters and, given its high DIC:TA ratio (>0.975), poses a potential threat to calcifying organisms. Upwelling of this $CO_2$-enriched ($pCO_2 > 600$ µatm) water to the surface

could change the net flux of $CO_2$ across the air-sea interface over the shelf, with possible outgassing of $CO_2$ or a suppression of the uptake of $CO_2$. The mean circulation through the two study months showed upwelling to be the dominant process, but changes in circulation seem to be common in this region, with both estuarine and anti-estuarine dynamics taking place.

It is important to look at how representative these two months are of other years, and of other times of the year as well.

Future monitoring should be carried out at a higher frequency (or continuously) to catch the system in its different modes of upwelling or downwelling. This would provide a more accurate picture of where nutrients, DIC, and metabolites are cycled with these strong current reversals. Models like the ANHA4 simulation used in this study can be used to forecast upwelling on the shelf and help determine how carbon cycling will be altered by these events. Combining our results with datasets

obtained from moorings and in-situ measurement platforms could improve estimates of carbon budgets in these traditionally hard to reach areas.

**6 Acknowledgements**

We thank the captain and crew of the CCGS *Amundsen* for their support and cooperation during field work. We are grateful to Pascal Guillot for the collection and processing of CTD data. We thank William Burt and Jonathan Lemay for help with sample analysis. This work was supported by the National Science and Engineering Research Council of Canada, ArcticNet, and GEOTRACES.

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

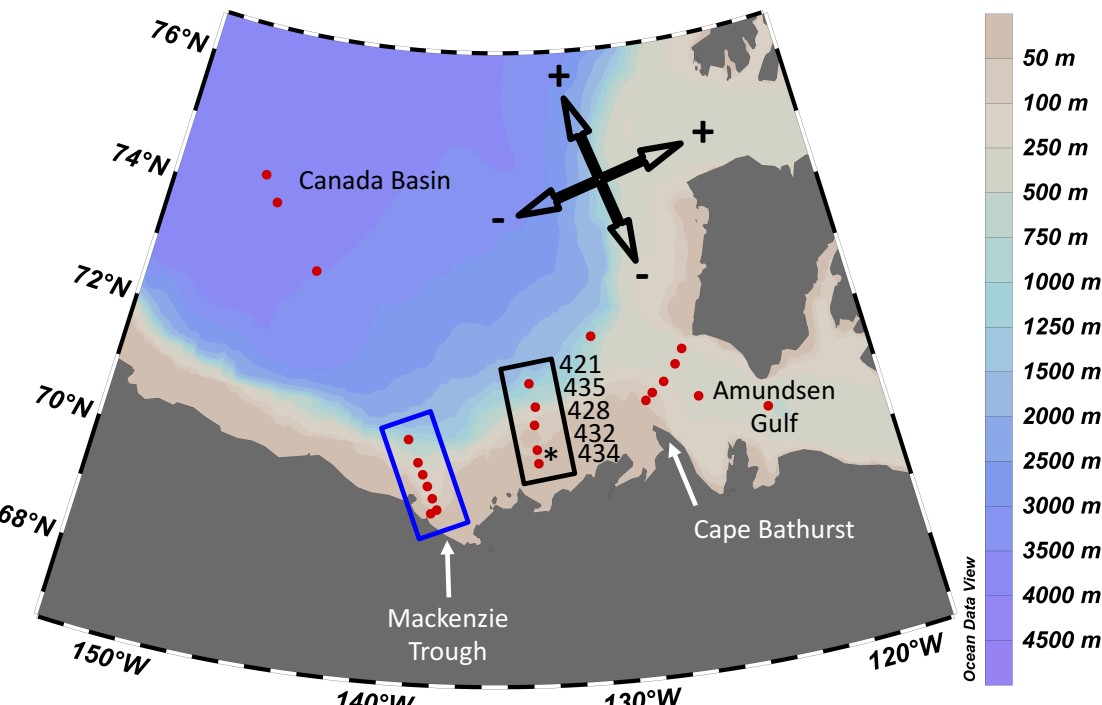

Figure 1: Map of stations (red dots) sampled in the Beaufort Sea during August and September of 2014. The black rectangle signifies the stations that make up the Mackenzie Shelf transect with numbers indicating station names. The blue rectangle signifies stations that make up the Mackenzie Trough transect. The black star in the Mackenzie Shelf transect indicates the location of wind data collection. The cross-shelf and along-shore directions are indicated by the black arrows.

**Table 1: End-member water mass properties used in the three component mass balance equations.**

| Water Mass | Salinity | $\delta^{18}$O-$H_2$O (‰) | TA ($\mu$mol kg$^{-1}$) | DIC ($\mu$mol kg$^{-1}$) | DIC/TA |
|---|---|---|---|---|---|
| MW | 0 | -19.1 | 1540 | 1390 | 0.90 |
| SIM | 4.7 | -2.0 | 415 | 330 | 0.80 |
| UHL | 33.1 | -1.4 | 2276 | 2226 | 0.98 |
| ATL | 34.8 | 0.18 | 2302 | 2167 | 0.94 |

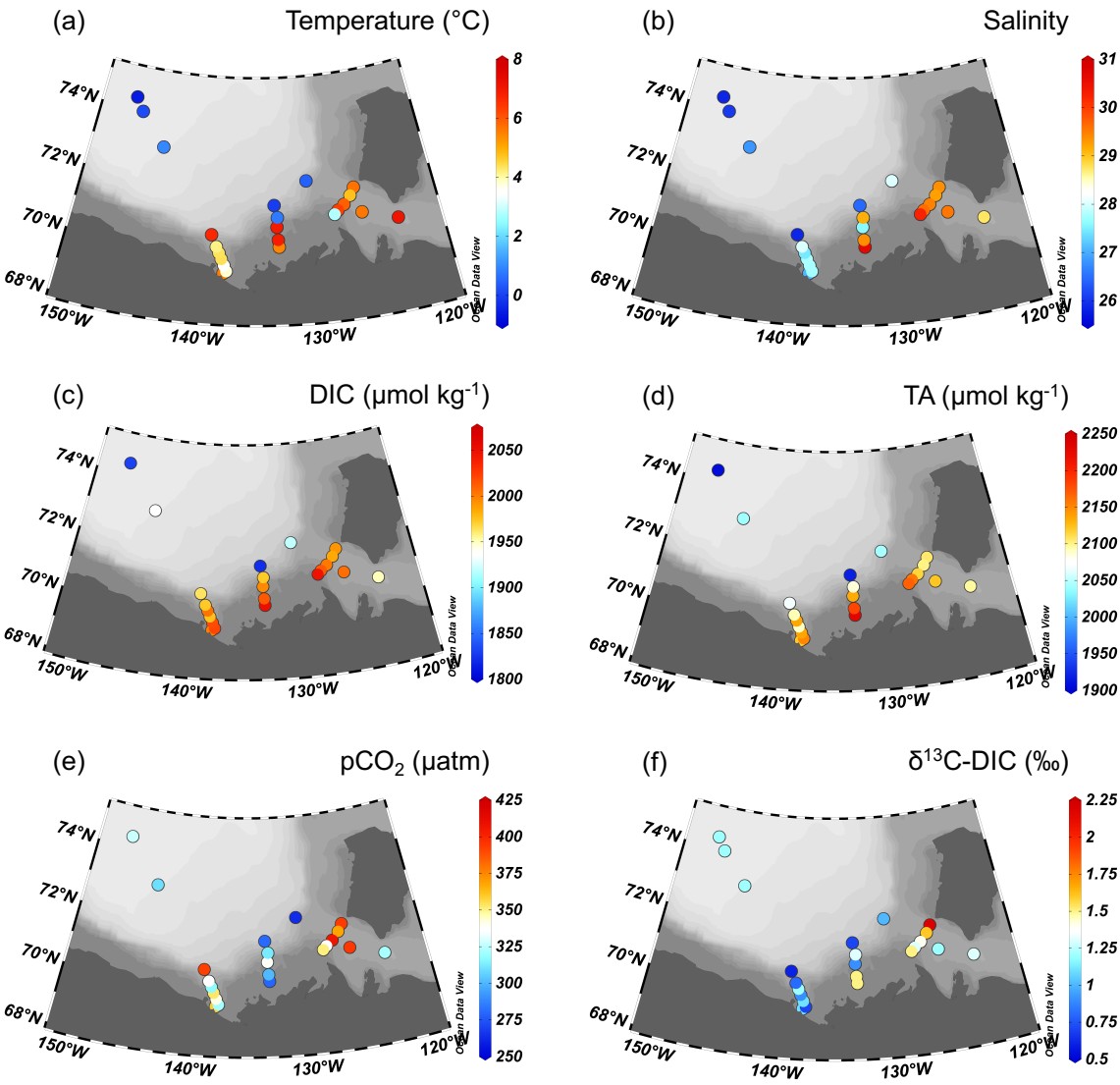

**Figure 2: A) Temperature, b) salinity, c) DIC, d) TA, e) *p*CO₂ and f) δ¹³C-DIC of surface samples in the study region measured at each station.**

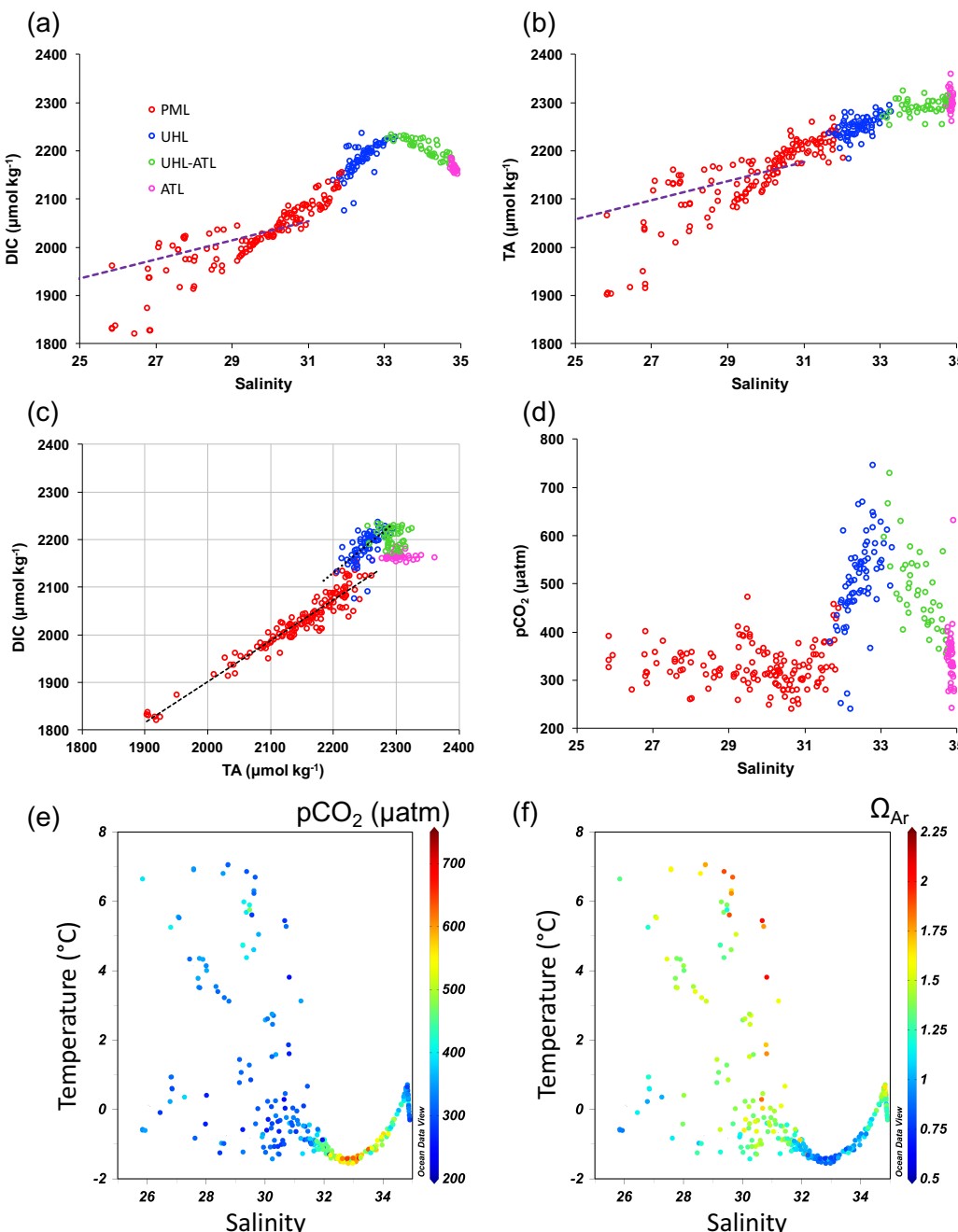

Figure 3: A) DIC versus salinity, b) TA versus salinity, c) DIC versus TA, d) $p$CO$_2$ versus salinity, e) temperature versus salinity with $p$CO$_2$, and f) temperature versus salinity with $\Omega_{Ar}$ for all samples at the stations shown in Fig. 1. Four water masses are identified by colour in figures a)-d): PML (red), UHL (blue), UHL-ATL (green) and ATL (pink). Dashed lines in a) and b) indicate linear regressions to a salinity of zero to find the Mackenzie River end-member of DIC and TA. Dashed lines in c) show the change in the relationship between DIC and TA in samples from the PML (red) and the UHL (blue). Slopes of these DIC:TA lines are 0.86 and 1.01 respectively.

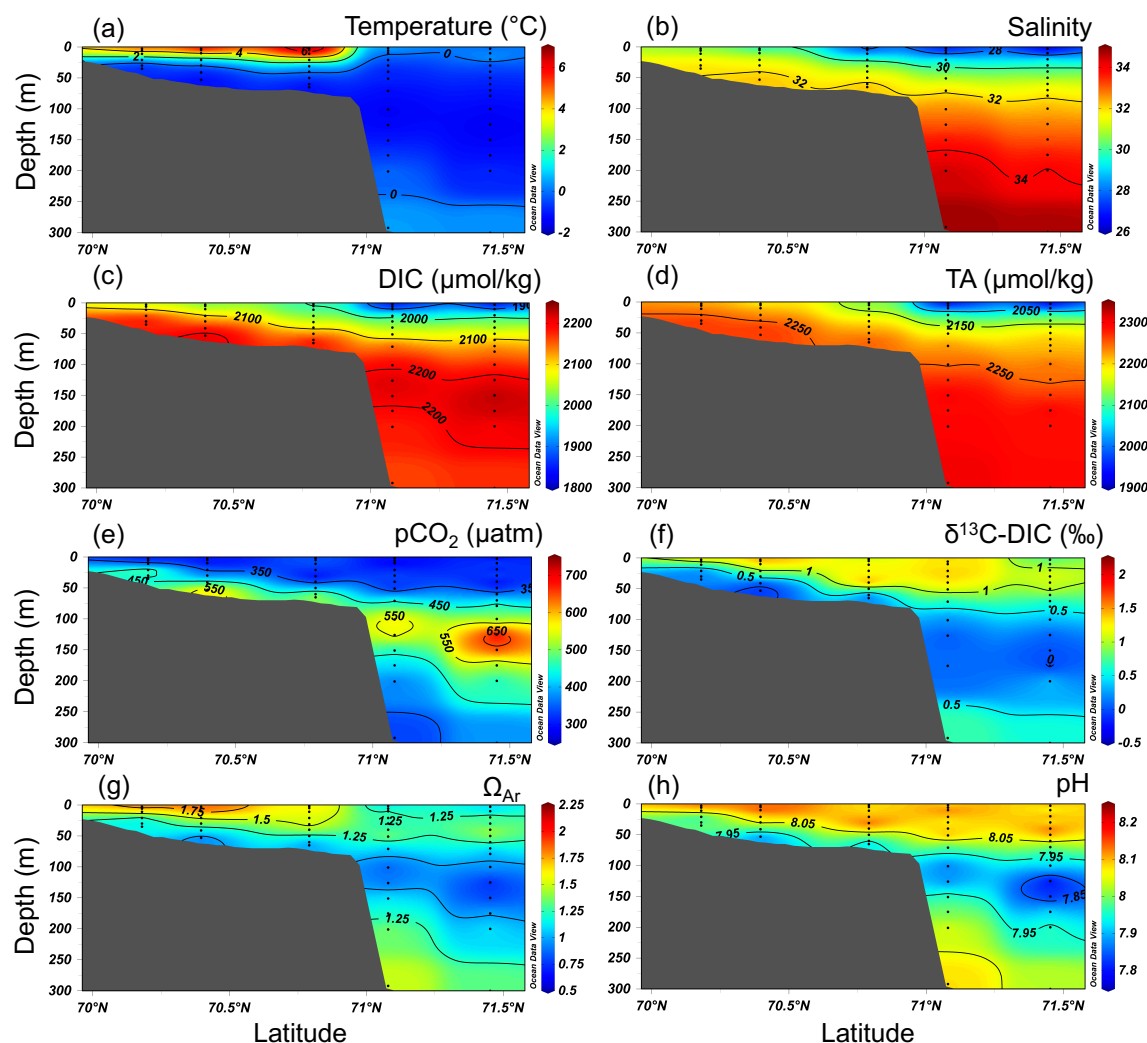

**Figure 4: Cross-sections of a) temperature, b) salinity, c) DIC, d) TA, e) $p$CO$_2$, f) δ$^{13}$C-DIC, g) Ω$_{Ar}$ and h) pH measured in, or computed for the top 300 m of the Mackenzie Shelf transect. Measurements are indicated by black dots.**

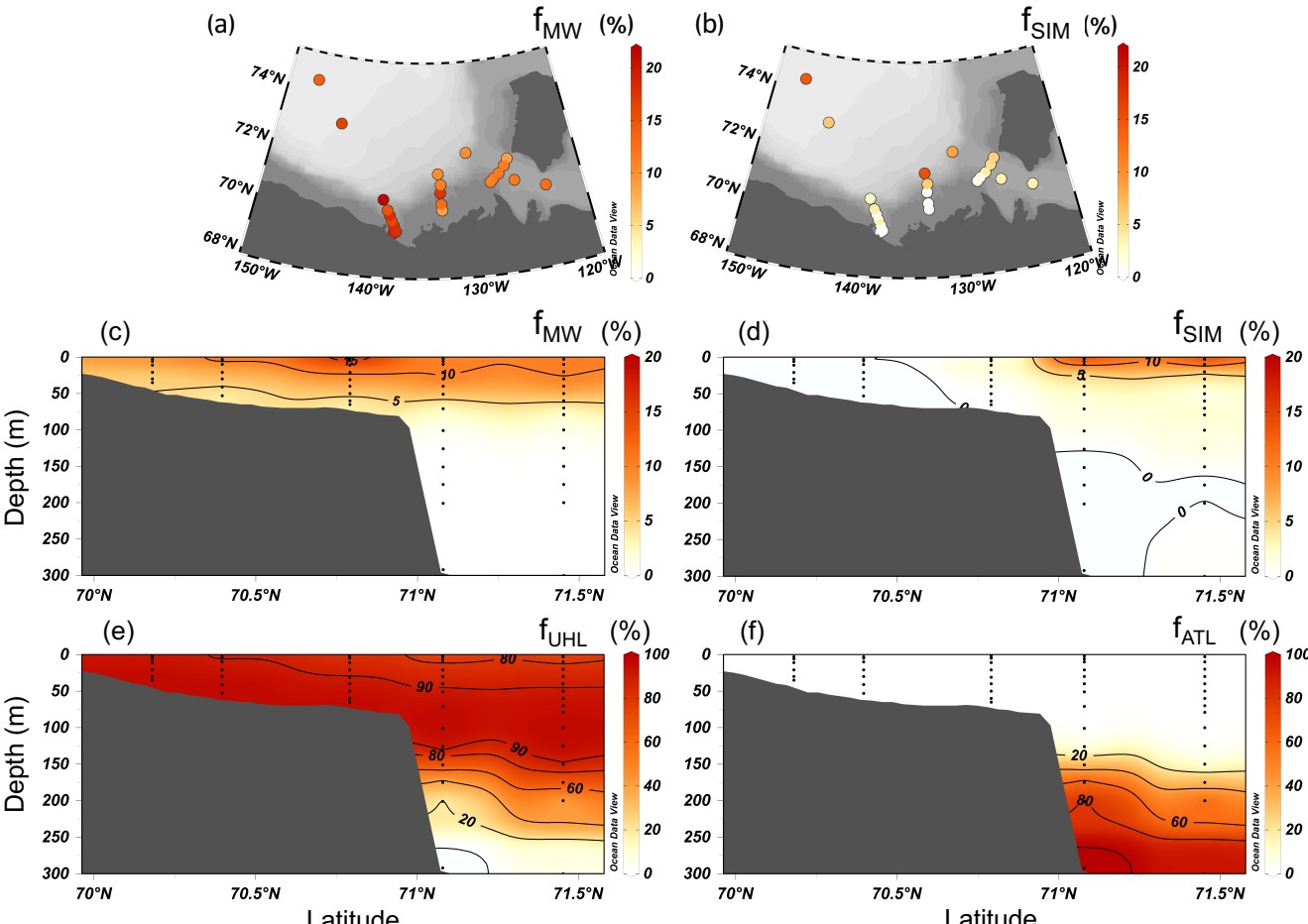

**Figure 5: Surface values of the fraction of a) meteoric water ($f_{MW}$) and b) sea-ice melt ($f_{SIM}$) at all stations in the study area. The fraction of c) meteoric water ($f_{MW}$), d) sea-ice melt ($f_{SIM}$), e) upper halocline layer ($f_{UHL}$) and f) Atlantic water ($f_{ATL}$) in the top 300 m of the Mackenzie Shelf transect. Measurements are indicated by black dots.**

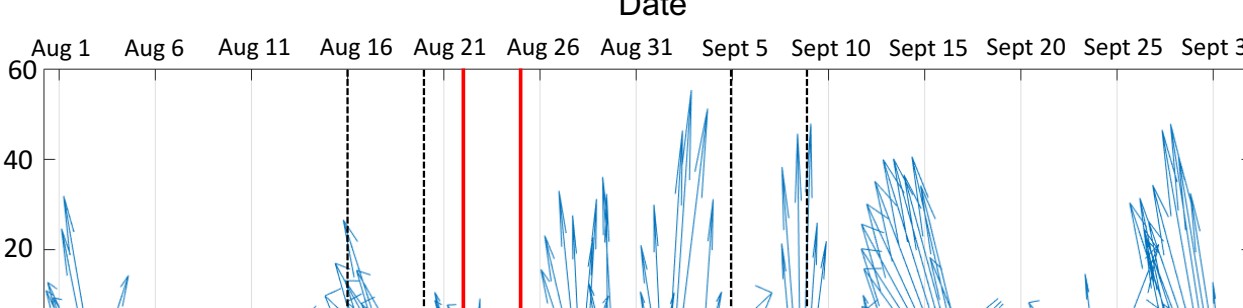

**Figure 6: 6-hour averaged winds for August and September of 2014 on the Mackenzie Shelf (location 70.31 °N, 133.59 °W) from gridded reforecast data from the Canadian Meteorological Centre. Red lines indicate the period of August 22nd to 24th in which sampling of the Mackenzie Shelf transect took place. Black dotted lines indicate the periods from August 16th to 20th when the greatest upwelling activity took place, and from September 5th to 9th when the greatest downwelling event occurred.**

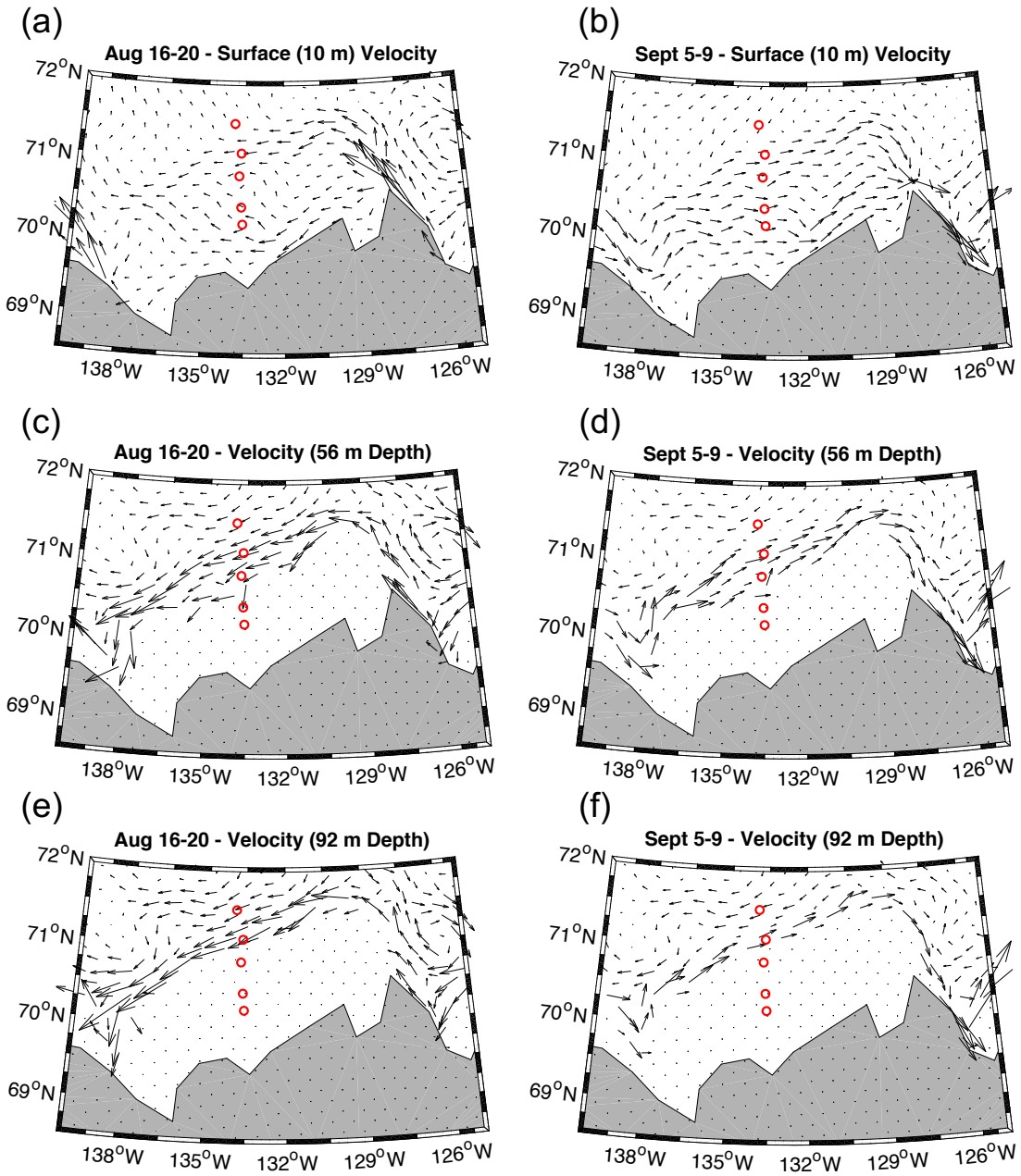

**Figure 7: Modelled velocity in the top 10 m of the water column (a and b), at 56 m depth (c and d), and at 92 m depth (e and f) on the Mackenzie Shelf during two time periods, from August 16th to 20th (a, c and e) and from September 5th to 9th (b, d and f) in the fall of 2014. Red circles indicate the stations sampled as part of the Mackenzie Shelf transect.**

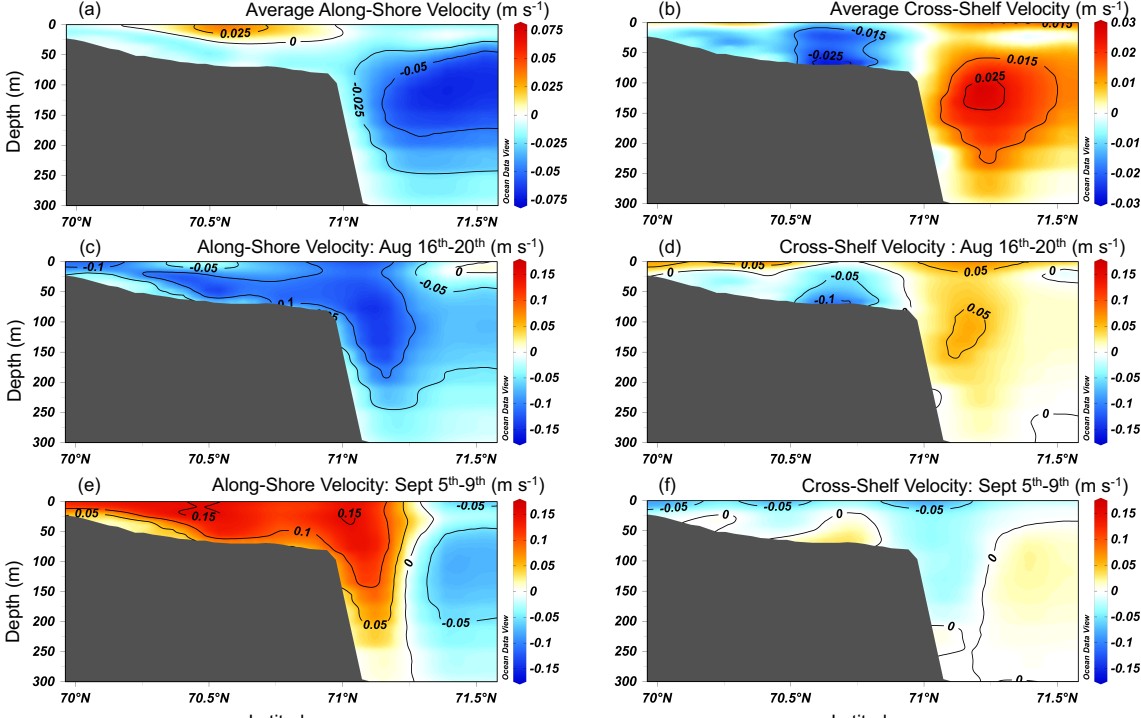

**Figure 8: The average a) along-shore and b) cross-shelf velocity on the Mackenzie Shelf transect during August and September of 2014, the c) along-shore and d) cross-shelf velocity from August 16th to 20th 2014, and the e) along-shore and f) cross-shelf velocity from September 5th to 9th. Positive values (red) indicate eastward along-shore flow (out of the plane) and off-shelf cross-shelf flow (to the right). Negative values (blue) indicate westward along-shore flow (into the plane) and on-shelf cross-shelf flow (to the left).**

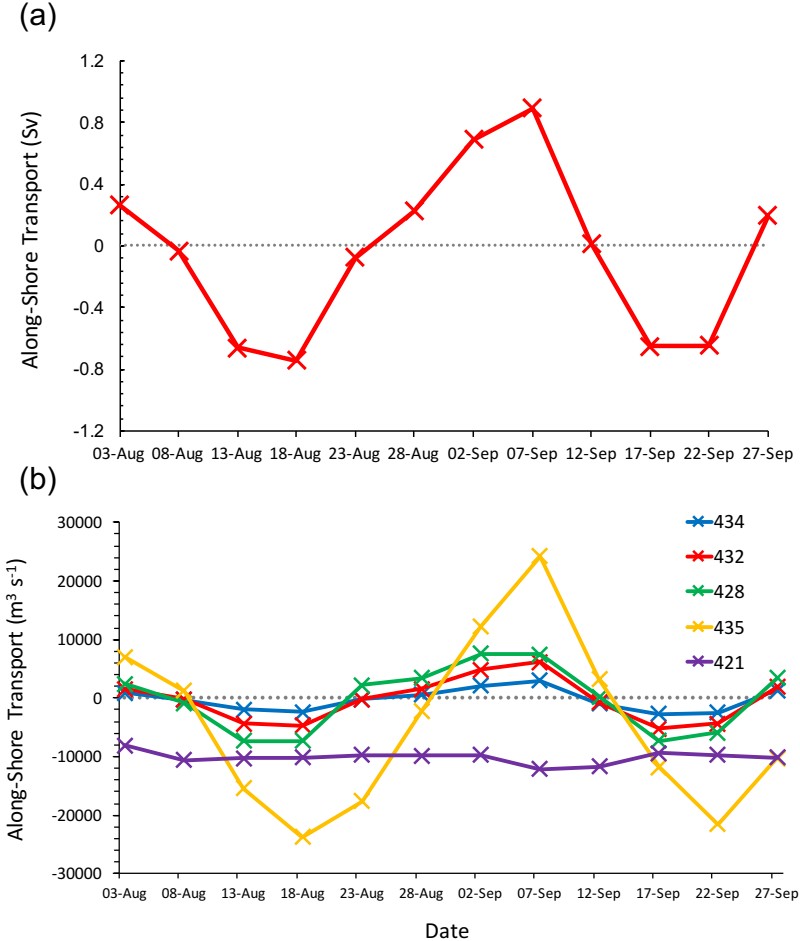

**Figure 9: The along-shore transport through the Mackenzie Shelf transect for the study period of August and September 2014. A) shows the total integrated volume transport for 120 km in the cross-shelf direction, avoiding the permanent westward transport of the Beaufort Gyre. Transport at each station, with a 1 km cross-shelf distance is shown in b). Transports are calculated from 5-day averages of modelled velocity.**

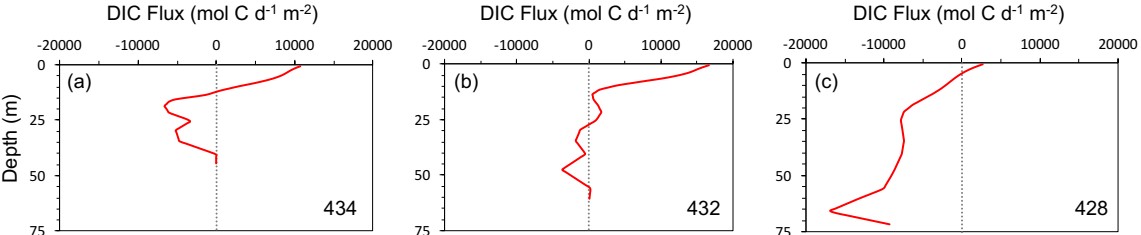

**Figure 10: Cross-shelf fluxes of DIC (mol C d$^{-1}$ m$^{-2}$) at stations a) 434, b) 432 and c) 428 on the Mackenzie Shelf. Positive values indicate off-shelf fluxes, negative values indicate on-shelf fluxes.**

**Table 2: Cross-shelf mass transports (Tg C d$^{-1}$) and excess carbon transport (relative to a $p$CO$_2$ of 400 µatm) calculated from measurements of DIC on the Mackenzie Shelf transect for the three shelf stations. The integrated mass transport is shown for both the top 10 m at each station and the full water column below the top 10 m with an along-shore distance of 10 km. Positive values indicate off-shelf transport; negative values indicate on-shelf transport.**

| Station | Water column section | Cross-shelf mass transport (Tg C d$^{-1}$) | Excess carbon transport (Tg C d$^{-1}$) |
|---------|---------------------|-------------------------------------------|-----------------------------------------|
| 434 | Top 10 m | $9.0 \pm 2 \times 10^{-3}$ | |
| | 10 – 45 m | $-14 \pm 3 \times 10^{-3}$ | $17 \pm 4 \times 10^{-5}$ |
| 432 | Top 10 m | $14 \pm 3 \times 10^{-3}$ | |
| | 10 – 61 m | $-3.0 \pm 3 \times 10^{-3}$ | $7.1 \pm 3 \times 10^{-5}$ |
| 428 | Top 10 m | $0.1 \pm 1 \times 10^{-3}$ | |
| | 10 – 72 m | $-65 \pm 15 \times 10^{-3}$ | $45 \pm 23 \times 10^{-5}$ |

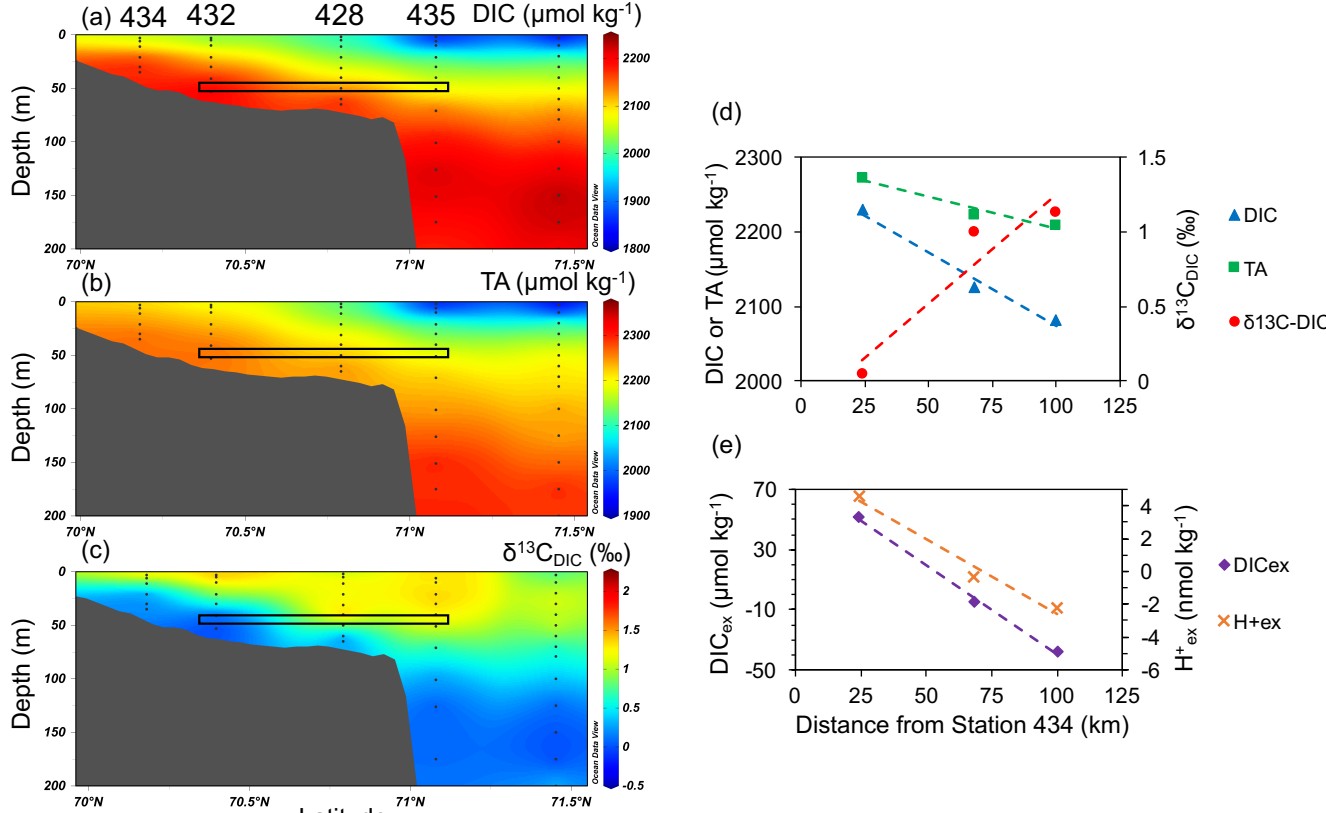

**Figure 11: Distributions of a) DIC, b) TA and c) δ$^{13}$C-DIC on the Mackenzie Shelf transect. Black boxes indicate depth of 50 m at stations 432, 428 and 435. Measurements are indicated by black dots. Chemical gradients at the 50-m depth horizon along the transect are shown in d) and e).**

