# Peer review of "Inorganic carbon fluxes on the Mackenzie Shelf of the Beaufort Sea"

_Biogeosciences, 2017_

## Referee Comment (RC1) · Anonymous Referee #1 · 6 Sep 2017

Overall Statements

The manuscript "Inorganic carbon fluxes on the Mackenzie Shelf of the Beaufort Sea" by J. Mol, H. Thomas, P.G. Myers, X. Hu, and A. Mucci presents the status of the carbonate system on a high latitude shelf sea in summer 2014. With the use of (surely intricate) measurements and a hydrodynamic model the authors describe the fluxes of the carbonate system relevant tracers. They discuss the significant consequences of an on-shore transport of low pH water masses.

The manuscript is well structured, is equipped with significant figures, and includes substantial conclusions. Despite a growing amount of flaws with higher page numbers I agree publishing the manuscript after minor revisions.

[Figure]

Detailed remarks

The wording "excess DIC" is used in excess. Please use it either for DIC/TA limits or for the Burt approach.

P4 L20 ff: Define UHL-ATL (see Fig. 3)

P4 L27: should be $\leq 33.1$

P4 L20 ff: The described layering of PML and UHL is a bit misleading when looking at Fig. 5, where the ratio of UHL water is on the shelf > 90 %

P6 L3: use "33 km horizontal .."

P6 L13: This transformation is hard to understand. What about a self-explaining sketch?

P7 L9: Which figure is meant?

P8 L 24: Give an explanation for the expected pattern.

P9 L7: MW fraction is still < 10 % there.

P9 L18: Show location in Fig. 1

P9 L25: "easterly"? I see rather north and south winds.

P9 L25: "wind speeds were comparable low"

P12 L8: You mean along-shelf here?

P12 L10: Fig. 9b

P12 L20: Fig. 10

P12 L20: per square meter: which along-shore distance was used?

P12 L32: in P8 L3 excess DIC/TA ratio was indicated for values > 1.

P13 L7 please refer here to Fig. 8d

P13 L9: Add "Table 2".

P13 L11: It should be mentioned that you use a "railway" approach for the transport of biogeochemical tracers. Please write something like: "Even though no results of a coupled physical – biogeochemical model was available the following arguments could be made .."

P13 L14 and L16: give start- and end-day.

P13 L13 Did you analyze the wind fields or the hydrodynamic results? Please add: "From the analysis of .., there are two .."

P13 L22: Fig. 11d

P13 L24 Two times "km"

P13 L31: Fig. 11c

P14 L10: Fig. 11e

P14 L16: "Table 2 last column"

P14 L25: Use space "ofCaCO3"

P20 Display the Amundsen Gulf and use white color for text within the continent.

P25: Also indicate 16-20 August and 5-9 September.

---

## Referee Comment (RC2) · Anonymous Referee #2 · 29 Sep 2017

This manuscript reports on carbonate system conditions on the Mackenzie Shelf of the Beaufort Sea during the summer of 2014. A three-component mixing model is used to estimate proportions of TA and $\delta18O$ originating from meteoric and riverine water, sea ice and Pacific water and this is cast within the context of a working circulation model (NEMO). They seek to understand the processes involved that regulate carbonate parameters on the shelf's mixed layer with the goal of quantifying shelf export. They find that upwelling, which was present prior to the cruise, and subsequent downwelling play important roles in controlling carbonate parameters.

The manuscript is well written and logically builds upon a growing body of observations and methods that have employed for interpretation of processes on the Arctic Shelves. They certainly did a lot with a sparse data set, which was taken during one cruise.

[Figure]

Nevertheless, using their data, prior literature values and models allows a reasonable means to interpret the dataset.

I recommend a revision addressing prior comments as well as those below.

1) The manuscript becomes poorly organized starting with the results. Some of what is reported in there (e.g. Wind Forcing 3.3) should probably go in a background section.

2) Pg. 2, L 16. The comment that, "N flux from the MacKenzie has little impact on NPP" should be cited.

3) Pg. 2, L 16. What happens to DON/PON? Is it totally refractory? A citation would help

4) Pg 6 L10-15. A diagram would be useful for the reader.

5) Pg 6 "Calculation uncertainty": Please clarify measurement uncertainty and methods. It's confusing. Where exactly did the standard deviations come from? How do you know the sensor or instrument errors? Also your approach requires that the errors be normally distributed. Is this the case?

6) Advertise your main findings in the abstract and quantify them in the conclusions. A casual reader will not get the results of your efforts.

7) Several other edits picked up by reviewer 1 definitely need to be addressed.

---

## Referee Comment (RC3) · Anonymous Referee #3 · 11 Oct 2017

The manuscript of "Inorganic carbon fluxes on the Mackenzie Shelf of the Beaufort Sea" by Jocoba Mol et al. reports the distribution of carbonate parameters on the Mackenzie shelf with data collected on a summer cruise in 2014, aiming to investigate the cycling of inorganic carbon. The authors conclude that wind forces controlled upwelling and downwelling play the key factors on regulating the inshore/offshore transport of inorganic carbon in both the surface mixed layer and the subsurface layer. In addition, the authors quantify the shelf export and further discuss the potential impacts of inorganic carbon transport on ocean acidification.

The manuscript is well structured and this work is solid although it is just coming from one cruise data. With the working circulation model, the authors successfully connect the potential influence of onshore/offshore carbon transport to ocean acidification.

[Figure]

However, the authors focus more on the mechanism controlling the inorganic carbon flux and the onshore/offshore transport over the shelf break only, rather than on the inorganic carbon flux coming from all sources. High paper numbers might bury the highlights while the manuscript could be further well organized starting from results.

I recommend a revision addressing the comments listed below and comments picked up by other reviewers. 1) It must be better to list major findings and quantified numbers in the abstract. 2) P4L27, UHL water has a salinity range, it's usually defined with salinities centered at S=33.1. 3) P4L28, Atlantic water is characterized by a temperature maximum while the salinity is not the largest in the salinity depth profile. Please define these water masses either by ranges of salinity, temperature, and or potential density, as well as depth range. 4) P4L30, have you considered the influence of denitrification on the conservation of TA? And how much error it will introduce into your calculation? 5) "p" in pCO2 is italic. 6) P5L15, where are the values of DIC from for these endmembers in Table 1? 7) P6L20, could you please list the uncertainties? 8) P6L30, please mark "Amundsen Gulf" in fig. 1. Not all readers know where it is. 9) P7L28, for fMW and fSIM, "MW" and "SIM" should be subscripted. 10) P7L34, "TA is conservative" is not really true. Given the authors discuss biological production, photosynthesis/respiration will change TA. In addition, denitrification also changes TA and pH. 11) P9L14, use period "break; it did". 12) P9L21, how long the wind should sustain to introduce upwelling or upwell subsurface water into the surface water on the Mackenzie shelf? 13) P11L32, Fig.10 is wrong. 14) P12L18, where is the flux of TA? 15) P13L18, how the bottom water impact the air-sea exchange? Is it more related to wind speed or the air-sea pCO2 difference? 16) P14L7, what's the importance of H+ flux? How is it related to DIC flux? Linear or exponential? 17) P14, p4.4, how low the aragonite saturation will impact the calcifier? And how low the saturation you observed? list the numbers. 18) Fig.11, list stations 428 and 435.

---

## Author Comment (AC1) · 8 Nov 2017

The authors would like to thank referee #1 for their insightful comments and helpful revisions of this manuscript. The authors agree for the most part with all of the revisions that have been suggested. Each of the referee's comments are individually addressed below, with the comment from referee #1 listed and then responded to.

The wording "excess DIC" is used in excess. Please use it either for DIC/TA limits or for the Burt approach.

Authors agree that the term "excess DIC" should only be used to address DIC/TA limits or for the Burt approach and will go through the manuscript to make this term more robust as a descriptor in these situations.

P4 L20 ff: Define UHL-ATL (see Fig. 3)

UHL-ATL will be defined in this paragraph, as it is used in the figure mentioned here. "Waters with a mixture of UHL and ATL properties, or without at least 80% of the total fraction of either water mass, are defined as UHL-ATL waters."

P4 L27: should be Âă33.1

This is the salinity that is mentioned at P4 L27, and it is the salinity seen at the UHL maximum.

P4 L20 ff: The described layering of PML and UHL is a bit misleading when looking at Fig. 5, where the ratio of UHL water is on the shelf > 90 %

It should be mentioned that the layering described here is more applicable over the deep basin and disappears somewhat over the shelf due to mixing/upwelling.

P6 L3: use "33 km horizontal .."

Adding horizontal to this statement does increase understanding.

P6 L13: This transformation is hard to understand. What about a self-explaining sketch?

It is possible to add a small direction figure onto the map in Fig 1 to display this transformation.

P7 L9: Which figure is meant?

Changing this line to "These surface plots (Fig. 2)..." would fix this.

P8 L 24: Give an explanation for the expected pattern.

These lines could be changed to explain the expected pattern with an addition such as "... in the surface or subsurface, due to photosynthesis and the preferential uptake of the lighter carbon isotope, and a decrease to minimum values of -0.5 per mil in the DIC maximum layer in the deep basin and on the shelf bottom due to respiration and the
release of these lighter carbon isotopes".

P9 L7: MW fraction is still < 10 % there.

Although the MW fraction is still quite high (>10%), this is the area where it is lowest. The relative fraction may be lower because of this intrusion of sea-ice melt and so it may be better to highlight the fact that there is a greater amount of sea-ice melt rather than saying that river input is more limited.

P9 L18: Show location in Fig. 1

The location where wind data was taken from can be shown on Fig 1.

P9 L25: "easterly"? I see rather north and south winds.

It may be more accurate to state that there were "both northeasterly and southeasterly winds throughout the beginning of the month."

P9 L25: "wind speeds were comparable low"

It should be added that "wind speeds were comparably low".

P12 L8: You mean along-shelf here?

There is a mistake in the cross/along wording in this paragraph. From P12 L8 it should read, "Along-shore transports at the stations on the transect are calculated through the water column, to a maximum depth of 200 m, by taking the sum of the products of the along-shore modelled velocity and a cross-shelf distance of 1 km (Fig 9b)".

P12 L10: Fig. 9b

This is correct, altered in the correction to this sentence above.

P12 L20: Fig. 10

This change to the figure number is correct.

P12 L20: per square meter: which along-shore distance was used?
An along-shore distance of 10 km was used in this calculation. This information should be added at P12 L19 to enhance understanding. "... through the water column with an along-shore distance of 10 km."

P12 L32: in P8 L3 excess DIC/TA ratio was indicated for values > 1.

It is mentioned at P8 L3 that a "greater ratio of DIC/TA indicates excess DIC", although this is not exactly equated with a value > 1. The value of 1 is indicating the slope of the line in Fig 3c. The wording at P8 L3 could be modified to further explain what is meant by this linear relationship, but also that the individual water masses have different DIC/TA ratios as shown in Table 1. None of these water masses actually have an average DIC/TA ratio > 1.

P13 L7 please refer here to Fig. 8d

This can be changed to "As expected by the cross-shelf velocity in Figure 8d, the transport in the top 10 m at all three stations is off-shelf..."

P13 L9: Add "Table 2".

Adding the reference to Table 2 at the end of P13 L9 would be helpful to the reader.

P13 L11: It should be mentioned that you use a "railway" approach for the transport of biogeochemical tracers. Please write something like: "Even though no results of a coupled physical – biogeochemical model was available the following arguments could be made .."

The beginning of this paragraph can be altered to read: "From August 6th to 16th, strong northeasterly winds were present over the shelf region, but had shifted to an easterly direction by the time sampling took place and through August 22nd. Although no results from a coupled physical-biogeochemical model were available, it seems probable from the wind and physical sampling data that upwelling on the Mackenzie Shelf began around August 6th and continued through to the 22nd of August..."

BGD
P13 L14 and L16: give start- and end-day.

Can change to "lasting longer than 10 days in August and September of 2014 (approximately August 10th to 28th and September 14th to 29th), during which..."

P13 L13 Did you analyze the wind fields or the hydrodynamic results? Please add: "From the analysis of .., there are two .."

Can alter this sentence to read: "From the analysis of the hydrodynamic model, there are two periods..."

P13 L22: Fig. 11d

This reference to Fig. 11d is correct.

```
P13 L24 Two times "km"
```

Not sure what this comment is related to. It might be advisable to remove the first set of units in this line and only keep the second after the second gradient.

P13 L31: Fig. 11c

The correction of this figure reference is correct.

P14 L10: Fig. 11e

The correction of this figure reference is correct.

P14 L16: "Table 2 last column"

This line should be changed to "...shown in the last column of Table 2".

P14 L25: Use space "ofCaCO3"

Should read "of CaCO3".

P20 Display the Amundsen Gulf and use white color for text within the continent.

A label of the Amundsen Gulf can be added and the text color will be changed to white
within the continent.

P25: Also indicate 16-20 August and 5-9 September.

Lines of another type can be added to indicated the time periods of 16-20 August and 5-9 September to indicate the times of strongest upwelling and downwelling behavior.

---

## Author Comment (AC2) · 8 Nov 2017

The authors wish to thank referee #2 for their insightful comments and helpful revisions of this manuscript. Each of the referee's comments are individually addressed below, with the comment first listed and then responded to.

1) The manuscript becomes poorly organized starting with the results. Some of what is reported in there (e.g. Wind Forcing 3.3) should probably go in a background section.

As this comment suggests a background section could be introduced, taking some of the results including the wind forcing and possibly water mass composition. Reorganizing these sections somewhat was considered, but ultimately left this way as a choice of style.

2) Pg. 2, L 16. The comment that, "N flux from the MacKenzie has little impact on NPP" should be cited.

Can be cited from the Tremblay et al., 2014 paper.

3) Pg. 2, L 16. What happens to DON/PON? Is it totally refractory? A citation would help

The authors mention this nitrogen dynamics only as a way to highlight the importance of upwelling for this shelf region with high river input. Since there is no actual mention of nutrient levels in the results or discussion, this section of the introduction was limited. It is possible to expand more on these nutrient dynamics, but seems somewhat unimportant to the main findings of the paper.

4) Pg 6 L10-15. A diagram would be useful for the reader.

As mentioned in reviewer 1's comments, a diagram can be added to Figure 1 to help clarify.

5) Pg 6 "Calculation uncertainty": Please clarify measurement uncertainty and methods. It's confusing. Where exactly did the standard deviations come from? How do you know the sensor or instrument errors? Also your approach requires that the errors be normally distributed. Is this the case?

We can change this paragraph to add more description of methods for finding the uncertainty. For this purpose, I believe we can assume that the errors are normally distributed. The effect of the errors of temperature, salinity and DIC are very small and do not give an uncertainty that is near the significance of the estimated transport. Most of the uncertainty comes from the model and this is the best way that we have seen fit to estimate the error coming from the ANHA4 model. "Uncertainty was estimated using Monte Carlo simulations for the calculation of cross-shelf mass transports. The inputs for the simulations were randomly generated from normal distributions, and 1000 points were chosen randomly for each of the variables. Standard deviation for salinity

(+/- 0.0003) and temperature (+/- 0.001 degC) were taken from error calculations with the Seabird SBE 911plus CTD sensor, and standard deviations for DIC measurements (+/- 2 umol/kg) were taken directly from repeat measurements of sample and standard measurements using the VINDTA 3C. The error of the modelled velocity was estimated by taking the standard deviation of the velocity at each station and depth layer from the ANHA4 model over the two-month study period."

6) Advertise your main findings in the abstract and quantify them in the conclusions. A casual reader will not get the results of your efforts.

The authors agree that the abstract and conclusion can be altered to more clearly address findings.

Abstract: P1L16: "A strong upwelling event prior to sampling on the Mackenzie Shelf took place, with high pCO2 water from the upper halocline layer (UHL) being moved onto the shelf bottom, with a maximum on-shelf DIC flux of 16.9 x 103 mol C d-1 m-2 during the event. The total on-shelf transport of carbon through the upwelling event was found to be 65 +/- 15 x 10-3 Tg C d-1. TA and the oxygen isotope ratio of water ($\delta$18O) are used to examine water-mass distributions in the study area and to investigate the influence of Pacific Water, Mackenzie River freshwater, and sea-ice melt on carbon dynamics and air-sea fluxes of carbon dioxide (CO2) in the surface mixed layer. Understanding carbon transfer in this seasonally dynamic environment is key to quantify the importance of Arctic shelf regions to the global carbon cycle and provide a basis for understanding how it will respond to the aforementioned climate induced changes."

Conclusion: P15L4: "Carbonate system parameters were measured in the Beaufort Sea during August and September of 2014. The cross-shelf transport of inorganic carbon during an upwelling event was estimated using the velocity fields from output of the ANHA4 simulation. This upwelling period from August 16th to 20th displayed an off-shelf transport in the surface layer of 14 +/- 3 x 10-3 Tg C d-1 and a corresponding

on-shelf transport in the subsurface of 65 +/- 15 x 10-3 Tg C d-1, bringing water from the UHL onto the shelf. This upwelled UHL water alters the carbonate chemistry of bottom waters and poses a potential threat to calcifying organisms due to the high DIC:TA ratio (> 0.975) present. Upwelling of this water with pCO2 concentrations > 600 uatm could change the net flux of CO2 across the air-sea interface over the shelf, with possible outgassing of CO2 or a suppression of the uptake of CO2. The mean circulation through the two study months showed upwelling to be the dominant process, but changes in circulation seem to be common in this region, with both estuarine and anti-estuarine dynamics taking place."

7) Several other edits picked up by reviewer 1 definitely need to be addressed.

See comments on reviewer 1's edits.

---

## Author Comment (AC3) · 8 Nov 2017

The authors wish to thank referee #3 for their insightful comments and helpful revisions of this manuscript. The authors agree for the most part with all of the revisions that have been suggested. Each of the referee's comments are individually addressed below, with the referee's comment listed first and subsequently responded to.

1) It must be better to list major findings and quantified numbers in the abstract.

See above response to reviewer 2's comment on abstract and conclusion (#6).

2) P4L27, UHL water has a salinity range, it's usually defined with salinities centered at S=33.1.

This should be changed to "a salinity range centered at 33.1".

3) P4L28, Atlantic water is characterized by a temperature maximum while the salinity is not the largest in the salinity depth profile. Please define these water masses either by ranges of salinity, temperature, and or potential density, as well as depth range.

The definitions of these water masses can be altered to read "The upper halocline layer (UHL) originates in the Pacific Ocean and lies below the PML. It covers a depth range of  $\sim 25$  to 150 m, a salinity range of 31.6 to 34.6 and is characterized by a temperature minimum. Below the UHL is the Atlantic layer (ATL), covering the depths below 150 m, with a salinity range of 34.6 to 34.9.

4) P4L30, have you considered the influence of denitrification on the conservation of TA? And how much error it will introduce into your calculation?

This hasn't been quantified for the water mass calculations. One reason for this is that the water mass fractions are only used as an identification tool and so any error from changing values of TA will not have a large impact on the outcome of the results.

5) "p" in pCO2 is italic.

The author will edit this point throughout the article.

6) P5L15, where are the values of DIC from for these endmembers in Table 1?

P5L20 add "The value of DIC for MW is taken from Shadwick et al. (2011b)." P5L20 "the values of TA, DIC and salinity for SIM are taken from Lansard et al. (2012). Values of del180, TA and DIC for the UHL are the average..."

Also need to change citation on P7L30, "These values are close to the Mackenzie River properties published in Shadwick et al. (2011b) and Cooper et al. (2008) used for the water mass decomposition..."

7) P6L20, could you please list the uncertainties?

BGD
See response to reviewer 2's fifth specific comment.

8) P6L30, please mark "Amundsen Gulf" in fig. 1. Not all readers know where it is.

As mentioned in the comments to reviewer 1, this label will be added to Figure 1.

9) P7L28, for fMW and fSIM, "MW" and "SIM" should be subscripted.

Making the MW and SIM subscripted would have to be altered throughout the paper and figures for MW, SIM, ATL and UHL.

10) P7L34, "TA is conservative" is not really true. Given the authors discuss biological production, photosynthesis/respiration will change TA. In addition, denitrification also changes TA and pH.

TA does change with biological activity, but not nearly as greatly as DIC. The sentence in question could be altered to better display that as something like, "TA is near conservative while DIC..."

11) P9L14, use period "break; it did".

Can be changed to "beyond the shelf break. It did not intrude..."

12) P9L21, how long the wind should sustain to introduce upwelling or upwell subsurface water into the surface water on the Mackenzie shelf?

This is a question that in order to give a real response I believe would require a more in depth look at the physics on the shelf and possibly running a model with directed wind. We can make an estimate of how long by looking at the hydrodynamic velocity through a time when wind was causing upwelling and the size of the shelf.

13) P11L32, Fig.10 is wrong.

Should be changed to Fig. 9.

14) P12L18, where is the flux of TA?

BGD
If this comment is in reference to Fig 10, which displays the DIC flux, if the TA flux is added onto this figure, it looks basically identical to the line of the DIC flux. Because of the magnitude of the fluxes, the lines do not differ enough to display them in this way for the UHL layer, which is the area of interest.

15) P13L18, how the bottom water impact the air-sea exchange? Is it more related to wind speed or the air-sea pCO2 difference?

The bottom water here impacts the air-sea exchange because of the air-sea pCO2 difference. This sentence could be changed to reflect that more closely: "... may impact the air-sea exchange of CO2 due to the large air-sea pCO2 difference including..."

16) P14L7, what's the importance of H+ flux? How is it related to DIC flux? Linear or exponential?

The H+ flux in this case is displaying the change in pH conditions along the shelf. It is positively related to the DIC flux across the shelf. The relationship is linear in this case, but is complicated by TA concentrations in certain areas.

17) P14, p4.4, how low the aragonite saturation will impact the calcifier? And how low the saturation you observed? list the numbers.

P14L25: Can state that the aragonite saturation in this water mass reaches values as low as 0.83. Varying impacts of these levels of aragonite saturation have been found within different communities and with different organisms. Could insert this type of a statement with reference to Ries et al., 2011 paper.

18) Fig.11, list stations 428 and 435.

This can be added to the figure to aid in station location.